# Continual Release Moment Estimation with Differential Privacy

**Nikita P. Kalinin**
Institute of Science and Technology Austria (ISTA)
Klosterneuburg, Austria
nikita.kalinin@ist.ac.at

**Jalaj Upadhyay**
Department of Computer Science
Rutgers University
Piscataway, NJ 08854, USA
jalaj.upadhyay@rutgers.edu

**Christoph H. Lampert**
Institute of Science and Technology Austria (ISTA)
Klosterneuburg, Austria
chl@ist.ac.at

## Abstract

We propose *Joint Moment Estimation* (JME), a method for continually and privately estimating both the first and second moments of a data stream with reduced noise compared to naive approaches. JME supports the *matrix mechanism* and exploits a joint sensitivity analysis to identify a privacy regime in which the second-moment estimation incurs no additional privacy cost, thereby improving accuracy while maintaining privacy. We demonstrate JME's effectiveness in two applications: estimating the running mean and covariance matrix for Gaussian density estimation and model training with DP-Adam.

## 1 Introduction

Estimating the first and second moments of data is a fundamental step in many machine learning algorithms, ranging from foundational methods, such as linear regression, principal component analysis, or Gaussian model fitting, to state-of-the-art neural network components, such as BatchNorm [23], and optimizers, such as Adam [32]. Often, these estimates must be computed and updated continuously, called the *continual release* setting [13]. For example, this is the case when the data arrives sequentially and intermediate results are needed without delay, such as for sequential optimization algorithms like Adam, or for real-time systems in healthcare, finance, or recommendation systems. In such scenarios, ensuring the privacy of sensitive data, such as user information or medical records, is essential.

*Differential Privacy (DP)* is a widely established formal notion of privacy that ensures that the inclusion or exclusion of any single individual's data has a limited impact on the output of an algorithm. Technically, DP masks sensitive information by adding suitably scaled noise, thereby creating a trade-off between privacy (formalized as a *privacy budget*) and utility (measured by the expected accuracy of the estimates). This trade-off becomes particularly challenging when more than one quantity is meant to be estimated from the same private data, as is the case when estimating multiple data moments. Done naively, the privacy budget has to be split between the estimates, resulting in more noise and lower accuracy for both of them.

In this work, we introduce a new method, *Joint Moment Estimation (JME)*, that is able to privately estimate the first and second moments of vector-valued data without suffering from this shortcoming. Algorithmically, JME relies on the recent *matrix factorization (MF)* mechanism [35] for continual

39th Conference on Neural Information Processing Systems (NeurIPS 2025).

Table 1: Common workload matrices for moment estimation in (1).

| method | symbol | coefficients |
|---|---|---|
| prefix sum | $E_1$ | $\mathbb{1}\{i \leq t\}$ |
| exponential (weight $\beta$) | $E_\beta$ | $\beta^{t-i}\mathbb{1}\{i \leq t\}$ |
| standard average | $V$ | $\frac{1}{t}\mathbb{1}\{i \leq t\}$ |
| sliding window (size $k$) | $W_k$ | $\frac{1}{k}\mathbb{1}\{t - k < i \leq t\}$ |

release DP to individually compute the first and the second moments of the data, thereby making it flexible to accommodate a variety of settings. For example, besides the standard uniform sum or weighted average across data items, exponentially weighted averages or sliding-window estimates are readily possible. The key innovation of JME lies in our theoretical analysis of its properties. By jointly analyzing the *sensitivity* of the otherwise independent estimation processes, in combination with considering a carefully calibrated trade-off between them, we show that one can estimate privately the second-moment matrix *without having to increase the amount of noise required to keep the first moment private.* In this sense, we obtain privacy of the second moment *for free*.

JME is practical and easily implemented using standard programming languages and toolboxes. We demonstrate this by showcasing two applications, one classical and one modern. First, we use JME for continual density estimation using a multivariate Gaussian model, e.g. we estimate the running mean and covariance matrix of a sequence of vector-valued observations. Our experiments confirm that JME achieves a lower Frobenius norm error for the covariance matrix in high-privacy regimes as well as a better fit to the true distribution as measured by the Kullback-Leibler divergence. Second, we integrate JME into the *Adam* optimizer, which is widely used in deep learning. Here, as well, we observe that JME achieves better optimization accuracy than baseline methods in high-privacy and small-batch-size regimes.

## 2   Background

We study the problem of differentially private continually estimation of the pair of weighted sums of the first and second moments, see Section 6 for a discussion of related works. Specifically, consider a sequence of $d$-dimensional vectors $x_1, \ldots, x_n \in \mathcal{X}$, where $\mathcal{X} = \{(x_1, \ldots, x_n) : x_i \in \mathbb{R}^d \wedge \max_i \|x_i\|_2 \leq \zeta\}$ for some fixed constant $\zeta > 0$. At each step $t$, we aim to privately estimate the following pair of sums:

$$Y_t = \sum_{i=1}^t a_i^t x_i \in \mathbb{R}^d \;\; \text{and} \;\; S_t = \sum_{i=1}^t b_i^t x_i x_i^\top \in \mathbb{R}^{d \times d}, \tag{1}$$

for arbitrary coefficients $a_i^t \in \mathbb{R}$ and $b_i^t \in \mathbb{R}$. This formulation includes many practical schemes; see Table 1. To express this problem compactly, we rewrite it in matrix form. We collect the coefficients into lower triangular workload matrices, $A_1 = (a_i^t)_{1 \leq i,t \leq n} \in \mathbb{R}^{n \times n}$ and $A_2 = (b_i^t)_{1 \leq i,t \leq n} \in \mathbb{R}^{n \times n}$. We stack the data as rows into a matrix $X \in \mathbb{R}^{n \times d}$. Then, all terms of the sums in (1) can then be expressed compactly[1] as $Y = A_1 X$ and $S = A_2(X \bullet X)$, where the second matrix consists of the data vectors stacked as rows, and $\bullet$ denotes the *Face-Splitting Product* or *transposed Khatri-Rao product* [16], $(X \bullet X)_{ij} := (x_i \otimes x_i)_j = \text{vec}(x_i x_i^\top)_j$, where $\otimes$ is the *Kronecker product* of two vectors.

To protect the privacy of individual data elements, we rely on the notion of *differential privacy*, considering neighboring datasets as those differing by a single data point; see e.g., [14] for an introduction. Recently, a series of works have developed effective methods for privately estimating individual matrices in the aforementioned product form by means of the *matrix mechanism* [35, 7, 10, 9, 12, 27, 22, 20, 17, 21, 8, 25, 39]. Its core insight is that, in the continual release setting, outputs at later steps should require less noise to be made private than earlier ones because more data points contributed to their computation. One can exploit this fact by adding noise that is *correlated* between the steps instead of being independent. This way, at each step, some of the noise added at an earlier stage can be removed again, thereby resulting in a less noisy result without lowering the privacy guarantees.

---

[1]We slightly abuse the notation and make $S$ a matrix of shape $n \times d^2$ instead of a tensor of size $n \times d \times d$.

---

**Algorithm 1** Joint Moment Estimation (JME)

---

**Require:** stream of vectors $x_1, \ldots, x_n \in \mathbb{R}^d$ with $\|x_t\|_2 \leq \zeta$,
**Require:** workload matrices $A_1, A_2 \in \mathbb{R}^{n \times n}$
**Require:** privacy parameters $(\epsilon, \delta)$
**Require:** noise shaping matrices $C_1, C_2 \in \mathbb{R}^{n \times n}$ (invertible, decreasing column norm), defaults: Id

$\quad \sigma_{\epsilon,\delta} \leftarrow$ noise strength for $(\epsilon, \delta)$-DP Gaussian mechanism
$\quad \lambda \leftarrow \|C_1\|_{1 \to 2}^2 / (c_d \zeta^2 \|C_2\|_{1 \to 2}^2)$ with $c_1 = \frac{8}{11 + 5\sqrt{5}}$, and $c_d = 2$ for $d \geq 2$      // scaling parameter
$\quad s \leftarrow 2\zeta \|C_1\|_{1 \to 2}$                                                                  // joint sensitivity
$\quad Z_1 \sim \left[\mathcal{N}(0, \sigma_{\epsilon,\delta}^2 s^2)\right]^{n \times d}, Z_2 \sim \left[\mathcal{N}(0, \sigma_{\epsilon,\delta}^2 s^2)\right]^{n \times d^2}$      // 1st and 2nd moment noise
$\quad$ **for** $t = 1, 2, \ldots, n$ **do**
$\quad\quad \widehat{x}_t \leftarrow x_t + [C_1^{-1} Z_1]_{[t, \cdot]}$ and $\widehat{x_t \otimes x_t} \leftarrow x_t \otimes x_t + \lambda^{-1/2} [C_2^{-1} Z_2]_{[t, \cdot, \cdot]}$
$\quad\quad$ **yield** $\quad \widehat{Y}_t = \sum\limits_{i=1}^{t} [A_1]_{t,i} \widehat{x}_i, \quad \widehat{S}_t = \sum\limits_{i=1}^{t} [A_2]_{t,i} \widehat{x_i \otimes x_i}$
$\quad$ **end for**

---

Technically, the matrix mechanism relies on an invertible *noise shaping matrix* $C$. For our theoretical results, we often also assume that $C$ has decreasing column norms, As the private estimate of a product $AX$, one computes $\widehat{AX} = A(X + C^{-1}Z)$, where $Z$ is a matrix of i.i.d. Gaussian noise. This estimate is $(\epsilon, \delta)$-differentially private if the noise magnitude is at least $\text{sens}(CX) \cdot \sigma_{\epsilon,\delta}$, where $\sigma_{\epsilon,\delta}$ denotes the variance required in the Gaussian distribution to ensure $(\epsilon, \delta)$-DP for sensitivity 1 queries[2]. $\text{sens}(\cdot)$ denotes the *sensitivity*, which, for any function $F$, is defined as

$$\text{sens}(F) = \max_{X \sim X'} \|F(X) - F(X')\|_F \tag{2}$$

where $X \sim X'$ denote two neighboring data matrices, i.e. identical except for one of the data vectors.

## 3 Joint Moment Estimation (JME)

We now introduce our main algorithmic contribution: the Joint Moment Estimation (JME) algorithm for solving the problem of differentially private (weighted) moment estimation in a continual release setting. Algorithm 1 shows its pseudo-code. JME takes as input a stream of input data vectors, the weights for the desired estimate in the form of two lower triangular workload matrices, and privacy parameters $\epsilon, \delta > 0$. Optionally, it takes two noise-shaping matrices as input. If these are not provided, $C_1 = C_2 = I_{n \times n}$, can serve as defaults.

The algorithm uses the matrix mechanism to privately estimate the weighted sums of vectors for both the first and second moments. At each step, the algorithm receives a new data point, $x_t$, it creates private version of both $x_t$ and $x_t \otimes x_t$ by adding suitably scaled Gaussian noise. The noise shaping matrices, if provided, determine the covariance structure of the noise.

To balance between the estimates of the first moment and of the second moment, a scaling parameter, $\lambda$, is used. In Algorithm 1, this parameter is fixed such that the total sensitivity of estimating both moments is equal to the sensitivity of just estimating the first moment alone. This implies that the overall noise variance to make both moment estimates private is equal to that needed to achieve the same level of privacy for the first moment. Consequently, the fact that we also privately estimate the second moment does not increase the necessary noise level for the first moment, a property we call **(second moment) privacy for free**.

Note that *privacy for free* is a quite remarkable property of JME. Generally, when using the same data more than once for private computations, more noise for each of them is required to ensure the overall privacy, following, e.g., the *composition theorems of DP* [26]. In some situations, however, one might also prefer a more flexible way to trade-off between the two moment estimates. For this, we present $\lambda$-JME in the appendix (Algorithm 3), a variant of JME that has $\lambda$ as a free hyper-parameter.

---

[2]Here and in the following we do not provide a formula for $\sigma_{\epsilon,\delta}$, because closed-form expressions have been shown to be suboptimal in some regimes. Instead, we recommend determining its value numerically, such as in [5]. For reference, typical values of $\sigma_{\epsilon,\delta}$ lie between approximately 0.5 (low privacy, e.g. $\epsilon = 8, \delta = 10^{-3}$) and 50 (high privacy, e.g. $\epsilon = 0.1, \delta = 10^{-9}$).

**Properties of JME** For some inputs $X = (x_1, \ldots, x_n) \in \mathbb{R}^{n \times d}$, let $Y = (Y_1, \ldots, Y_n) \in \mathbb{R}^{n \times d}$ and $S = (S_1, \ldots, S_n) \in \mathbb{R}^{n \times d^2}$ be the matrices of their first and second moments after each step, and let $\widehat{Y} = (Y_1, \ldots, Y_n) \in \mathbb{R}^{n \times d}$ and $\widehat{S} = (S_1, \ldots, S_n) \in \mathbb{R}^{n \times d^2}$ be the private estimates computed by Algorithm 1 with workload matrix $A$ and noise shaping matrix $C$. Then, we have the following.

**Theorem 3.1** (Noise properties of JME). *$\widehat{Y}$ and $\widehat{S}$ are unbiased estimates of $Y$ and $S$. Their estimation noise, $\widehat{Y} - Y$ and $\widehat{S} - S$, has a symmetric distribution.*

**Theorem 3.2** (Privacy of JME). *Algorithm 1 is $(\epsilon, \delta)$-differentially private.*

**Theorem 3.3** (Utility of JME). *For any input $X$, let $c_d$ be as defined in Algorithm 1. Then the expected approximation error of Algorithm 1 for the first and second moments are*

$$\sqrt{\mathbb{E}\|Y - \widehat{Y}\|_F^2} = 2\zeta\sqrt{d}\sigma_{\epsilon,\delta}\|C_1\|_{1 \to 2}\|A_1 C_1^{-1}\|_F, \tag{3}$$

$$\sqrt{\mathbb{E}\|S - \widehat{S}\|_F^2} = 2\zeta^2\sqrt{c_d}d\sigma_{\epsilon,\delta}\|C_2\|_{1 \to 2}\|A_2 C_2^{-1}\|_F, \tag{4}$$

*where $\| \cdot \|_{1 \to 2}$ denotes the maximum $\ell_2$ column norm of the given matrix.*

*Proof of Theorem 3.1.* The statement follows from the fact that for any $t = 1, \ldots, n$, JME's estimates $\widehat{x_t}$ of $x_t$ and $\widehat{x_t \otimes x_t}$ of $x_t \otimes x_t$ are unbiased with symmetric noise distribution because they are constructed by adding zero-mean Gaussian noise. $\qquad\square$

*Proof sketch of Theorem 3.2.* The result follows using Gaussian mechanism once we show that the overall sensitivity of jointly estimating both moments has at most $s = 2\zeta\|C_1\|_{1 \to 2}$, as stated in Algorithm 1.

**Definition 3.4.** For any noise-shaping matrices $C_1$ and $C_2$ and any $\lambda > 0$, we define the *joint sensitivity* of estimating the first and $\lambda$-weighted second moment by

$$\begin{aligned} \text{sens}_\lambda^2(C_1, C_2) &= \sup_{X \sim X'} \left\| \begin{pmatrix} C_1 & \mathbf{0} \\ \mathbf{0} & \sqrt{\lambda}C_2 \end{pmatrix} \begin{pmatrix} X - X' \\ X \bullet X - X' \bullet X' \end{pmatrix} \right\|_F^2 \\ &= \sup_{X \sim X'} \left[ \|C_1(X - X')\|_F^2 + \lambda\|C_2(X \bullet X - X' \bullet X')\|_F^2 \right]. \end{aligned} \tag{5}$$

The following lemma (shown in the appendix) characterizes the values of the joint sensitivity as a function of $\lambda$.

**Lemma 3.5** (Joint Sensitivity). *Assume that the matrices $C_1$ and $C_2$ have norm-decreasing columns[3]. Then, for any $\lambda > 0$ holds:*

$$sens_\lambda^2(C_1, C_2) = \zeta^2\|C_1\|_{1 \to 2}^2 r_d \left( \frac{\lambda\zeta^2\|C_2\|_{1 \to 2}^2}{\|C_1\|_{1 \to 2}^2} \right), \tag{6}$$

*where $\| \cdot \|_{1 \to 2}$ denotes the maximum column norm, corresponding to the norm of the first column of $C_1$ and $C_2$, respectively. The function $r_d(\nu)$ is given by*

$$r_d(\nu) = \begin{cases} \frac{1}{8}(3 - \tau)^2(\nu\tau + 1 + \nu), \text{ with } \tau = \sqrt{1 - \frac{2}{\nu}}, & \text{if } \nu > \frac{11+5\sqrt{5}}{8}, d = 1, \\ 2 + 2\nu + \frac{1}{2\nu}, & \text{if } \nu > \frac{1}{2}, d > 1, \\ 4, & \text{otherwise.} \end{cases} \tag{7}$$

Algorithm 1 uses the scaling parameter $\lambda = \|C_1\|_{1 \to 2}^2/(c_d\zeta^2\|C_2\|_{1 \to 2}^2)$, so, by Lemma 3.5, the sensitivity of estimating both moment has the value $\sqrt{\zeta^2\|C_1\|_{1 \to 2}^2 r_d(c_d)}$. The choice of $c_d$ implies that $r_d(c_d) = 4$, which yields $\text{sens}_\lambda(C_1, C_2) = s$. This concludes the proof of Theorem 3.2. $\quad\square$

---

[3]The decreasing column norm structure ensures that the optimum is simultaneously achieved for the first columns of the matrices $C_1$ and $C_2$. This can be relaxed by solving a maximization problem over the column indices, which could be done numerically. Moreover, the matrices that we have in mind for practical tasks are Toeplitz, for which the norm-decreasing property is fulfilled naturally.

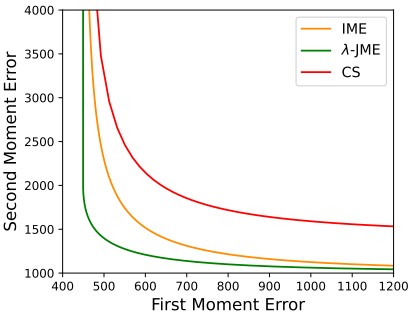

Figure 1: Approximation errors for the 1st and 2nd moments across three methods (see text for details): $\lambda$-JME with $\lambda$, IME with varying $\alpha$ parameters, and CS with varying $\tau$ parameter. $\lambda$-JME Pareto-dominates the other two methods.

*Proof sketch of Theorem 3.3.* The identities follow from the general properties of the matrix mechanism. For any input $X$, the output of Algorithm 1 for the first moment is $\widehat{Y} = Y + A_1 C_1^{-1} Z_1$, where $Z_1 \sim [\mathcal{N}(0, \sigma_{\epsilon,\delta}^2 s^2)]^{n \times d}$. Consequently, $\widehat{Y} - Y = A_1 C_1^{-1} Z_1$, and hence

$$\mathbb{E}_{Z_1} \|\widehat{Y} - Y\|_{\mathrm{F}}^2 = \|A_1 C_1^{-1}\|_{\mathrm{F}}^2 \cdot \sigma_{\epsilon,\delta}^2 s^2 d \tag{8}$$

Equation (3) follows by inserting $s = 2\zeta \|C_1\|_{1 \to 2}$. For the second moment, Equation (4) follows analogously using that the output of Algorithm 1 is $\widehat{S} = S + \lambda^{-\frac{1}{2}} A_2 C_2^{-1} Z_2$ with $Z_2 \sim [\mathcal{N}(0, \sigma_{\epsilon,\delta}^2 s^2)]^{n \times d^2}$. $\qquad\square$

### 3.1 Comparison with alternative techniques

Besides JME, alternatives methods are possible that could be used to privately estimate the first and second moments. In this section, we introduce some of them and discuss their relation to JME.

**Independent Moment Estimation (IME).** A straight-forward way to privately estimate first and second moments is to estimate and privatize both of them separately, where the necessary amount of noise is determined by the composition theorem of the Gaussian mechanism [1] (see Algorithm 4 in the appendix).

IME resembles JME in the sense that (i) separate estimates of the moments are created, and (ii) the privatized results are unbiased estimators. However, it does not have JME's *privacy for free* property. This is because IME relies on the composition theorem, so the privacy budget is split into two parts, one per estimate, where the exact split is a hyperparameter of the method. As a consequence, IME's estimate of the first moment is always more noisy, and thereby of lower expected accuracy, than JME's. For the second moment, IME could in principle achieve a lower noise than plain JME by adjusting the budget split parameter in an uneven way. This, however, would come at the expense of further increase in the error for estimating the first moment.

The following theorem establishes that $\lambda$**-JME, the variant of JME with adjustable $\lambda$ parameter offers a strictly better trade-off than IME.**

**Theorem 3.6** (JME vs IME). *For any $\epsilon, \delta > 0$, $\lambda$-JME Pareto-dominates IME with respect to the approximation error for the first vs second moment estimates.*

The proof can be found in the appendix. Figure 1 visualizes this property graphically.

**Concatenate-and-split (CS).** In the special case where the two noise shaping matrices are meant to be the same (e.g. the trivial case where both are the identity matrix), it is possible to use a single privatization step for both moments. For this, one forms a new observation vector, $\tilde{x}_i$ by concatenating $x_i$ with a vectorized (and potentially rescaled version of) $x_i \otimes x_i$. Then, one privatizes the resulting vector, taking into account that $\tilde{x}_i \in \mathbb{R}^{d(d+1)}$ has higher dimension and a larger norm than the original $x_i \in \mathbb{R}^d$. The result is split again into first and second order components, and the latter is unscaled. The result are private estimate of $x_t$ and $x_t \otimes x_t$, from which the two weighted moment estimates

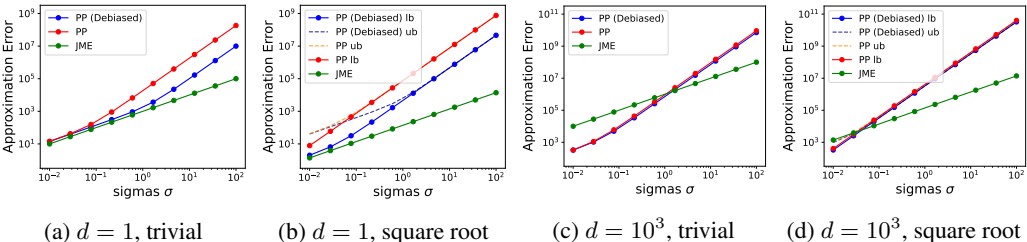

| (a) $d = 1$, trivial | (b) $d = 1$, square root | (c) $d = 10^3$, trivial | (d) $d = 10^3$, square root |

Figure 2: Expected error of second moment estimation with JME versus PP with and without debiasing ($A = E_1$ (prefix sum), $n = 1000$) In line with our analysis, for $d = 1$ JME consistently achieves a higher quality than PP. For $d = 1000$, JME is preferable to PP in the high privacy regime. Furthermore, the square root matrix factorization substantially improves the quality of both methods.

can be constructed. Algorithm 5 in the appendix provides pseudocode for this *concatenate-and-split (CS)* method.

When applicable, CS is easy to implement and produces unbiased moment estimates. However, like IME, it has an unfavorable privacy-accuracy trade-off curve compared to JME, as formalized in the following theorem and shown in the appendix.

**Theorem 3.7** (JME vs CS). *For any $\epsilon, \delta > 0$, $\lambda$-JME Pareto-dominates CS with respect to the approximation error for the first vs second moment estimates.*

Figure 1 visualizes these cases (Theorems 3.6 and 3.7) graphically in an exemplary setting ($d = 10, n = 100, C_1 = C_2 = I, A = E_1, \zeta = 1, \sigma_{\epsilon,\delta} = 1/2$).

**Post-processing (PP).** Post-processing (PP) is another easy-to-use method for joint moment estimation. It has appeared in the literature [40], at least in its naive form without the matrix factorization mechanism. For any $x_i$, it first computes a private estimate $\widehat{x}_t$ by adding sufficient noise to it. It then sets $\widehat{x_t \otimes x_t} := \widehat{x}_t \otimes \widehat{x}_t$, which is automatically private by the postprocessing property of DP, and uses it to estimate the second moment matrix without additional privacy protection. Algorithm 6 in the appendix provides pseudocode.

Like JME, PP has the *privacy-for-free* property, i.e., the private estimate of the second moment does not reduce the quality of the first moment. In contrast to JME, however, PP's estimate of the second moments is not unbiased because the noise that was added to the first moment is squared during the process. It is possible to compensate for this by explicitly subtracting the bias, which can be computed analytically. The bias depends only on hyperparameters such as $n$, $d$, and $\sigma$, and not on the private data $X$ or the sampled noise used to protect it. Its exact expression can be found in equation (60).

For given privacy parameters, PP and JME use the same noise strength toprivatize the first moment and therefore their estimates are of equal quality. For the second moment, their characteristics differ between the low privacy (small noise variance) and the high privacy (large noise variance) regime. To allow for a quantitative comparison, we derive characterizations of the approximation quality of PP.

**Lemma 3.8** (Expected Second Moment Error for PP). *Assume the same setting as for Theorem 3.3, except that we compute the estimates $\widehat{Y}$ and $\widehat{S}_{PP}$ with the PP method. Let $Q = C_1^{-1} C_1^{-\top}$ and $E_Q = \operatorname{diag}(Q) \operatorname{diag}(Q)^\top$. Then, the expected approximation error of the second moment satisfies:*

$$\sup_{X \in \mathcal{X}} \mathbb{E} \|S - \widehat{S}_{PP}\|_F^2 = 16d(d+1)\sigma_{\epsilon,\delta}^4 \zeta^4 \|C_1\|_{1\to2}^4 \operatorname{tr}(A_2^\top A_2 (Q \circ Q)) \tag{9}$$

$$+ 8(d+1)\sigma_{\epsilon,\delta}^2 \zeta^2 \|C_1\|_{1\to2}^2 \sup_{X \in \mathcal{X}} \operatorname{tr}((A_2^\top A_2 \circ Q) X X^T) \overbrace{+ 16d\sigma_{\epsilon,\delta}^4 \zeta^4 \|C_1\|_{1\to2}^4 \operatorname{tr}(A_2^\top A_2 E_Q)}^{bias}.$$

*For debiased PP, the term marked "bias" does not occur.*

In order to get a better impression of the relation between PP and JME, we first study the special case of a trivial factorization.

**Corollary 3.9.** *Assume that JME and PP use trivial factorizations, i.e., $C_1 = C_2 = I$. Then, JME's expected approximation error for the second moment is*

$$\mathbb{E} \|S - \widehat{S}\|_F^2 = 4c_d d^2 \sigma_{\epsilon,\delta}^2 \zeta^4 \|A_2\|_F^2 \tag{10}$$

*and the corresponding value for PP is*

$$\sup_{X \in \mathcal{X}} \mathbb{E}\|S - \widehat{S}_{PP}\|_F^2 = \left(16d(d+1)\zeta^4\sigma_{\epsilon,\delta}^4 + 8(d+1)\sigma_{\epsilon,\delta}^2\zeta^2\right)\|A_2\|_F^2 + \overbrace{16d\sigma_{\epsilon,\delta}^4\zeta^4\|\sum_i (A_2)_{[i,\cdot]}\|_2^2}^{bias},$$

*where $\sum_i (A_2)_{[i,\cdot]}$ is the row-wise summation of $A_2$. For debiased PP, the term marked "bias" does not occur.*

*Proof.* The proofs follow directly from Theorem 3.3 and Lemma 3.8 by observing that $\|C_1\|_{1\to 2} = \|C_2\|_{1\to 2} = 1$. In dimension $d = 1$, $\zeta = 1$, JME has an error of $4c_1\sigma_{\epsilon,\delta}^2\|A_2\|_F^2$ versus $(32\sigma_{\epsilon,\delta}^4 + 16\sigma_{\epsilon,\delta}^2)\|A_2\|_F^2$ for (debiased) PP. Since $c_1 < 1$, **for one-dimensional data, JME's error is always lower than PP's** (see Figure 2 for visual confirmation). In high dimensions ($d \gg 1$), the terms quadratic in $d$ dominate, so the comparison is between $8\sigma_{\epsilon,\delta}^2\|A_2\|_F^2$ for JME and $16d^2\sigma_{\epsilon,\delta}^4\|A_2\|_F^2$ for PP. Consequently, at least **for $\sigma_{\epsilon,\delta} \geq \frac{1}{\sqrt{2}}$, JME achieves privacy with less added noise than PP.**

In the regime of *low privacy* ($\sigma_{\epsilon,\delta} \to 0$) in high dimension ($d \gg 1$), PP can be expected to result in higher accuracy estimates than JME, because the terms involving $\sigma_{\epsilon,\delta}^4$ make only minor contributions.

For settings with general noise shaping matrix, we cannot provide an exact comparison between PP and JME, because the $\sup_X$-term in Equation (9) has no closed-form solution. Instead, we derive upper and lower bounds (Lemma D.2), and provide a numeric comparison in Figure 2.

# 4  Applications

To demonstrate potential uses of JME, we highlight two applications: *private Gaussian density estimation*, a classical probabilistic technique, and *private Adam optimization*, which is common in deep learning. The proofs for all theoretical results can be found in the appendix.

**Private Gaussian density estimation** The maximum likelihood solutions to Gaussian density estimation from i.i.d. data, $x_1, \ldots, x_t$ famously is $\widehat{p}(x) = \mathcal{N}(x; \mu_t, \Sigma_t)$, where $\mu_t = \frac{1}{t}\sum_{i=1}^t x_i$ is the *sample mean* and $\Sigma_t = \frac{1}{t}\sum_{i=1}^t (x_i - \mu_t)(x_i - \mu_t)^\top$ is the *sample covariance*. Using the alternative identity $\widehat{\Sigma}_t = \frac{1}{t}\sum_{i=1}^t x_i x_i^\top - \mu_t\mu_t^\top$, one sees that the task can indeed be solved in a continuous release setting and that only estimates of the first and second moments are required.

To do so privately, we use the *averaging* workload matrix $V = (a_i^t)$ with $a_i^t = \frac{1}{t}$ for $1 \leq i \leq t$ and $a_i^t = 0$ otherwise, and we compute private estimates $\widehat{\mu} := \widehat{VX}$ (private mean) and $\widehat{\Sigma} = \widehat{V(X \bullet X)} - \widehat{(VX)} \bullet \widehat{(VX)}$ (private covariance) using JME.

Note that $\widehat{\mu}$ is simply the first-moment vector as above. It is therefore unbiased and the guarantees of Theorem 3.3 holds for it. However, $\widehat{\Sigma}$ is not an unbiased estimate because in its computation the noise within $\widehat{VX}$ is squared. However, it is possible to characterize the bias analytically and subtract it if required. We provide the exact expression for the bias in equation (75) in the Appendix, where the debiased version is referred to as **JME (Debiased)**.

The expected approximation error of $\widehat{\mu}$ is identical to the one in Theorem 3.3 with $A = V$ and $C_1 = C_2 = I$. The following Theorem establishes the approximation quality of the covariance estimate. In this and the following sections we denote $\sigma = 2\sigma_{\epsilon,\delta}$, referring to the strength of the noise we add to the first moment for $\zeta = 1$ and trivial factorization.

**Theorem 4.1** (Private covariance matrix estimation with JME)**.** *Assume that all input vectors have norm at most 1. Let $\widehat{\Sigma}$ be the results of the above construction, where privacy is obtained by running JME with noise strength $\sigma$ and debiasing. Then it holds:*

$$\sup_{X \in \mathcal{X}} \mathbb{E}\|\Sigma - \widehat{\Sigma}\|_F^2 = (c_d d^2 + 2d + 2)\sigma^2 H_{n,1} + d(d+1)\sigma^4 H_{n,2}, \tag{11}$$

*with $c_d$ as in Theorem 3.3, and $H_{n,m} := \sum_{k=1}^n \frac{1}{k^m}$.*

For comparison, we also analyze the case where the PP method is used to privatize the covariance matrix, $\widehat{\Sigma}_{PP} := V(\widehat{X} \bullet \widehat{X}) - (V\widehat{X} \bullet V\widehat{X})$, where $\widehat{X}$ are privatized entries of $X$. Again, the resulting biased estimate can be explicitly debiased, see Appendix (93) for the expression.

The following theorem states upper and lower bounds on the expected approximation error:

**Theorem 4.2** (Private covariance matrix estimation with PP). *Assume the same setting as for Theorem 4.1. Let $\widehat{\Sigma}_{PP}$ be the result of the above construction, where privacy is obtained by running PP with noise strength $\sigma$ and debiasing. Let $S(n, d, \sigma) := d(d+1)\sigma^4 H_{n,1} - d(d+1)\sigma^4 H_{n,2} + 2(d+1)\sigma^2 H_{n,1}$. Then, for the expected error of the covariance matrix estimate it holds:*

$$S(n, d, \sigma) - 2(d+1)\sigma^2 H_{n,3} \leq \sup_{X \in \mathcal{X}} \mathbb{E} \|\Sigma - \widehat{\Sigma}_{PP}\|_F^2 \leq S(n, d, \sigma). \tag{12}$$

**Comparison.** In the high privacy regime ($\sigma \gg 1$), the leading term for JME is $d(d+1)H_{n,2}\sigma^4$, and for PP it is $d(d+1)H_{n,1}\sigma^4$. Given that $H_{n,1} = O(\log n)$, while $H_{n,m} < \pi^2/6$ for $m \geq 2$, the error introduced by PP is logarithmically worse than JME. Figure 6 shows a numerical plot that confirms this observation. Note that our results match the lower bounds established by G. Kamath 2020 [30], who proved that private covariance estimation in the Frobenius norm requires $\Omega(d^2)$ samples.

### Private Adam optimization

The *Adam* optimizer [32], has become a de-facto standard for optimization in deep learning. The defining property of Adam is its update rule, $\theta_i \leftarrow \theta_{i-1} - \alpha m_i/(\sqrt{v_i} + \epsilon)$, where $\alpha$ is a learning rate, and $m_i$ and $v_i$ are exponentially running averages of computed model gradients and componentwise squared model gradients, respectively. In the context of our work, these quantities correspond to a weighted first-moment vector and the diagonal of the weighted second-moment matrix.

Previous attempts to make Adam differentially private relied on postprocessing [4, 36], potentially with debiasing [42], i.e. they privatized the model gradients and derived the squared values from these. We demonstrate that JME's approach of privatizing both quantities separately can be a competitive alternative. Algorithm 2 in the Appendix shows the pseudocode.

It contains some modifications compared to the original JME. In particular, we adjust JME to only estimate and privatize the diagonal of the second moment matrix, which reduces the runtime and memory requirements. Interestingly, as the next theorem shows, having to estimate only the diagonal elements of the covariance matrix does not reduce the problem's sensitivity, so privatizing the estimates remains equally hard. This implies that our previous analyses, including the relation to the baselines, remain valid.

**Theorem 4.3.** *The sensitivity of JME, with the whole second-moment matrix $(C_1 X, \sqrt{\lambda} C_2 X \bullet X)$, and with just the diagonal terms $(C_1 X, \sqrt{\lambda} C_2 X \circ X)$, are identical.*

The proof can be found in the Appendix A.3. It does not follow from Lemma 3.5 and is significantly more intricate.

As in the previous cases, JME's *for free* property ensures that its expected approximation error of the first moment is identical to PP. The following theorem establishes the expected approximation errors of both methods for the computed second moments (i.e. the diagonal of the second moment matrix):

**Lemma 4.4** (Comparison of JME and PP for DP-Adam). *Let $D = (v_1, \ldots, v_n)$ be the matrix of second moment estimates of the Adam algorithm. Denote by $\widehat{D} = (\widehat{v}_1, \ldots, \widehat{v}_n)$ the private estimate of $D$ computed by Algorithm 2 with trivial factorization and noise strength $\sigma$, and let $\widehat{D}_{PP}$ be the analog quantity computed by DP-Adam with postprocessing. Then it holds:*

$$\mathbb{E}_Z \|\widehat{D} - D\|^2 = 2d\sigma^2 \cdot \|A_2\|_F^2, \tag{13}$$

$$\sup_{X \in \mathcal{X}} \mathbb{E}_Z \|\widehat{D}_{PP} - D\|^2 = (2d\sigma^4 + 4\sigma^2) \cdot \|A_2\|_F^2 \overbrace{+ d\sigma^4 \cdot \big\| \sum_i (A_2)_i \big\|_2^2}^{bias}, \tag{14}$$

*where $A_2$ is the workload matrix obtained from the coefficients of Adam's exponentially weighted averaging operations. The term marked "bias" disappears if PP is debiased.*

The proof of the first part follows directly from Theorem 3.3. The error for PP is computed separately in Lemma D.5 in the appendix.

This lemma shows that as long as $\sigma \geq 1$, the error introduced by JME is strictly lower than that of PP (i.e. classic DP-Adam [36]) and even the debiased version of PP (i.e. DP-AdamBC [42]). Figure 5 illustrates the relation in a prototypical case.

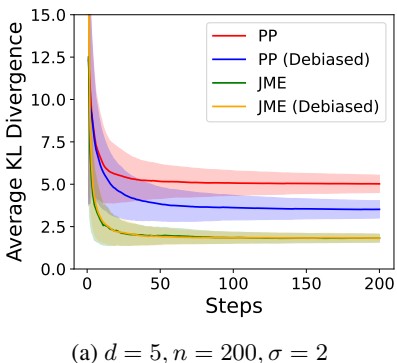
(a) $d = 5, n = 200, \sigma = 2$

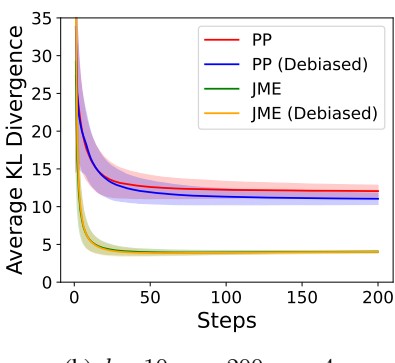
(b) $d = 10, n = 200, \sigma = 4$

Figure 3: Approximation quality of JME and PP for private Gaussian density estimation as average and standard deviation over 1000 runs. In the high-privacy regime ($\sigma = 2$ and $\sigma = 4$), JME achieves lower KL divergence than PP after $\approx 10$ samples.

## 5 Experiments

Our main contributions in this work are both algorithmic and theoretical. Specifically, JME is the general purpose technique for moment estimation, which is promising for some scenarios and less promising for others. However, it is also a *practical* algorithm that can be easily implemented and integrated into standard machine learning pipelines. To demonstrate this, we report on the experimental result of using Algorithm 1 and Algorithm 2 in two exemplary settings, reflecting the application scenarios described above.

**Private Gaussian density estimation.** From a given data distribution, $p(x) = \mathcal{N}(\mu, \Sigma)$, we sample $n = 200$ data points and use either JME or PP to form a private estimates, $\widehat{p}_t(x) = \mathcal{N}(\widehat{\mu}_t, \widehat{\Sigma}_t)$, at each step $t = 1, \ldots, n$, of the continuous release process. To ensure positive definiteness, we symmetrize JME's estimated covariance matrices and project them onto the positive definite cone. As a postprocessing operation, this does not affect their privacy.

Figure 3 shows the results, with the approximation quality, measured by the Kullback-Leibler (KL) divergence, $\text{KL}(\widehat{p}_t \| p)$, at each step, $t$, as average and standard deviation over 1000 runs, with $\mu \sim \mathcal{N}(0, \frac{1}{2}I_d)$ and $\Sigma \sim W_d\left(\frac{1}{2}I_d, 2d\right)$ (i.e. a *Wishart distribution*) in each case. One can see that in the high-privacy regime (here: $\sigma = 2$), on average, JME achieves a better estimate of the true density than PP, with and without debiasing.

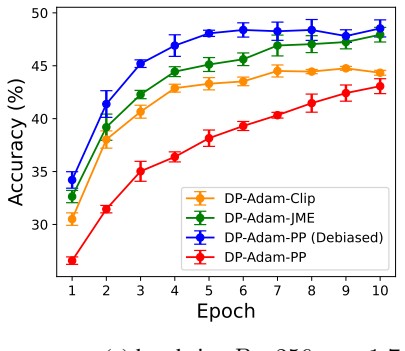
(a) batchsize $B = 256, \varepsilon \approx 1.7$

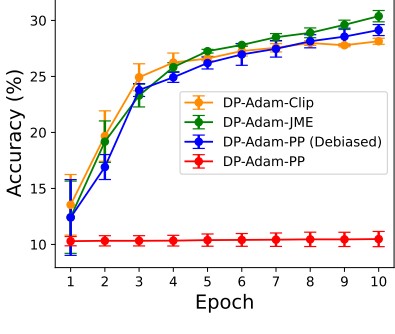
(b) batchsize $B = 1, \varepsilon \approx 0.16$

Figure 4: Results of private Adam training on CIFAR-10 experiments comparing four methods: DP-Adam based PP (with and without debiasing), JME, and joint clipping, over 10 epochs. Left: for low to medium privacy, JME outperforms vanilla DP-Adam but is slightly worse than the debiased version. Right: for high privacy, JME is slightly better than the debiased version, whereas vanilla DP-Adam cannot handle the necessarily amounts of noise.

**Private model training with Adam.** We train a convolutional network on the CIFAR-10 dataset with DP-Adam, which is privatized either with JME or PP. Because gradients could have arbitrarily large norm, we apply gradient clipping to a model-selected threshold in both methods. In addition, we include a heuristic baseline that uses the concatenate-and-split techniques in which not the norm of the gradient vector but the norm of the concatenation vector is clipped. While inferior to JME in the worst case, this might be beneficial for real data, so we include it in the experimental evalution.

The results in Figure 4 confirm our expectations: in a setting with small noise variance (large batchsize, low privacy), DP-Adam-JME achieved better results than standard DP-Adam, but worse than the debiased variant of DP-Adam. If the noise variance is large (small batchsize, high privacy), DP-Adam-JME slightly improves over the other methods. For detailed accuracy results, see Table 2 in the appendix with hyperparameters provided in Table 3.

## 6    Related works

The problem of differentially private moments (and the related problem of covariance estimation) has a rich history of development, with optimal results known in the central model of privacy [41] as well as the *local model of privacy* [11], and also in the worst-case setting. Sheffet [40] proposes three differentially private algorithms for approximating the second-moment matrix, each ensuring positive definiteness. The related setting of covariance estimation has been studied in the worst-case setting when the data comes from a bounded $\ell_2$ ball by several works for approximate differential privacy [2, 6, 15, 37, 38, 43] with the works of Amin *et al.* [3] and Kapralov & Talwar [31] presenting a pure differentially private algorithm. Covariance estimation is also used as a subroutine in mean-estimation work under various distributional assumptions [29]. However, none of these approaches are directly applicable to the continual release model and they offer improvements over the Gaussian mechanism only in a very high privacy regime. Precise sensitivity analysis, as a technique to improve differential privacy guarantees, has also been intensively used in the literature [34, 44, 24, 33]. The matrix mechanism has also gained a lot of attention recently due to its application of continually releasing prefix sums in private online optimization [8, 25, 39, 20, 17, 21, 28, 18, 19].

## 7    Summary and Discussion

In this work we studied the problem of jointly estimating the first and second moment of a continuous data stream in a differentially private way. We presented the Joint Moment Estimation (JME) method, which solves the problem by exploiting the recently proposed matrix mechanism with carefully tuned noise level. As a result, JME produces unbiased estimates of both moments while requiring less noise to be added than baseline methods, at least in the high privacy regime. We applied JME to private Gaussian density estimation and model training with Adam, demonstrating improved performance in high-privacy regimes both theoretically and practically.

Despite the promising results, several open question remain. In particular, we would like to explore if postprocessing is indeed the optimal strategy in a low-privacy regime, or if a better privacy-utility trade-off is still possible. Furthermore, we plan to explore the possibility of problem-specific factorizations, which could be fused with the proposed method.

## Acknowledgments

We thank Monika Henzinger for her valuable feedback and insightful discussions on earlier versions of this draft. We are also grateful to Mher Safaryan for his contributions to discussions on DP-Adam. Additionally, we thank Ryan McKenna for suggesting Joint Clipping as a baseline.

Jalaj Upadhyay's research was funded by the Rutgers Decanal Grant no. 302918, NSF CNS 2433628, Google Research Scholar Award, and Google Seed Fund Grant. A part of this work was done while visiting the Institute of Science and Technology Austria (ISTA).

Nikita Kalinin's research was funded in part by the Austrian Science Fund (FWF) [10.55776/COE12].

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

# A Proofs of the main theorems

In this section, we provide proofs of Theorem 3.2 and Theorem 3.3 from Section 3 as well as Theorem 4.3 from Section 4.

## A.1 Proof of Theorem 3.2 and Lemma 3.5

The privacy of Algorithm 1 follows from the properties of matrix mechanism [35], with a precise estimate of the *sensitivity* of the joint estimation process that we will introduce and discuss later in this section.

By means of the matrix factorization mechanism with $A_1 = B_1 C_1$ and $A_2 = B_2 C_2$, we write the joint moment estimate as,

$$\begin{pmatrix} Y & \mathbf{0} \\ \mathbf{0} & S \end{pmatrix} = \begin{pmatrix} B_1 & \mathbf{0} \\ \mathbf{0} & \lambda^{-\frac{1}{2}} B_2 \end{pmatrix} \begin{pmatrix} C_1 X & \mathbf{0} \\ \mathbf{0} & \lambda^{\frac{1}{2}} C_2 (X \bullet X) \end{pmatrix} \tag{15}$$

where $\lambda > 0$ is an arbitrary trade-off parameter that we will adjust optimally later. To make the estimate private, we privatize the rightmost matrix in (15), which contains the data, by adding suitably scaled Gaussian noise. The subsequent matrix multiplication then acts on private data, so its result is also private.

It remains to show that the noise level specified in Algorithm 1 suffices to guarantee $(\epsilon, \delta)$-privacy.

For this, we denote by $\mathcal{X} = \{(x_1, \ldots, x_n) \; : \; \max_i \|x_i\|_2 \leq \zeta \wedge x_i \in \mathbb{R}^d\}$ be the set of *input sequences* where each $d$-dimensional vector has bounded $\ell_2$ norm. The *sensitivity* of a computation is the maximal amount by which the result differs between two input sequences $X, X' \in \mathcal{X}$, which are identical except for a single data vector at an arbitrary index (denoted by $X \sim X'$).

For JME, the relevant sensitivity is the one of the matrix we want to privatize, i.e. (in the squared form)

$$\begin{aligned} \mathrm{sens}_\lambda^2(C_1, C_2) &= \sup_{X \sim X'} \left\| \begin{pmatrix} C_1 & \mathbf{0} \\ \mathbf{0} & \sqrt{\lambda} C_2 \end{pmatrix} \begin{pmatrix} X - X' \\ X \bullet X - X' \bullet X' \end{pmatrix} \right\|_\mathrm{F}^2 \\ &= \sup_{X \sim X'} \left[ \|C_1(X - X')\|_\mathrm{F}^2 + \lambda \|C_2(X \bullet X - X' \bullet X')\|_\mathrm{F}^2 \right] \end{aligned} \tag{16}$$

Due to the linearity of the operations and the condition imposed by $X \sim X'$, most terms in (16) cancel out and the value of $\mathrm{sens}_\lambda^2(C_1, C_2)$ simplifies into the solution of the following optimization problem.

**Problem 1** (Sensitivity for Joint Moment Estimation)**.**

$$\max_{i=1,\ldots,n} \max_{\substack{\|x\|_2 \leq \zeta \\ \|y\|_2 \leq \zeta}} \alpha_i^2 \|x - y\|_2^2 + \lambda \beta_i^2 \|x \otimes x - y \otimes y\|_2^2, \tag{17}$$

where $\alpha_i^2 = \|(C_1)_i\|_2^2$ and $\beta_i^2 = \|(C_2)_i\|_2^2$ are the column norms of the matrices $C_1$ and $C_2$, respectively.

To study Problem 1, we introduce as an intermediate object the formulation of (17) in the special case of $\zeta = 1$ and $\alpha_i = \beta_i = 1$ for $i = 1, \ldots, n$, treated as a function of $\lambda$:

$$r_d(\lambda) := \max_{\substack{x,y \in \mathbb{R}^d \\ \|x\|_2 \leq 1 \\ \|y\|_2 \leq 1}} \|x - y\|_2^2 + \lambda \|x \otimes x - y \otimes y\|_\mathrm{F}^2. \tag{18}$$

The following theorems provide specific values for $r_d$ in the special case of $d = 1$ (Theorem A.1) and for general $d \geq 2$ (Theorem A.2):

**Theorem A.1** ($d = 1$)**.** *For any $\lambda > 0$, it holds:*

$$r_1(\lambda) = \begin{cases} 4 & \text{if } \lambda \leq \frac{11 + 5\sqrt{5}}{8}, \\ \frac{1}{8}(3 - \tau)^2(\lambda\tau + 1 + \lambda) & \text{otherwise.} \end{cases} \tag{19}$$

*where $\tau = \sqrt{1 - 2/\lambda}$. Moreover, the function $r_1(\lambda)$ is a continuous function with respect to the parameter $\lambda > 0$.*

**Theorem A.2** (Joint Sensitivity for Moments Estimation). *For $d > 1$ and $\lambda > 0$:*

$$r_d(\lambda) = \begin{cases} 4 & \text{if } \lambda \leq \frac{1}{2}, \\ 2 + 2\lambda + \frac{1}{2\lambda} & \text{otherwise.} \end{cases} \tag{20}$$

The proofs of both theorems can be found further in the appendix. Theorem A.1 requires only straightforward optimization. For theorem A.2, we rewrite the Frobenius norm of the difference of Kronecker products and optimize over all possible values for $\langle x, y \rangle$.

As a corollary of Theorems A.1 and A.2, we obtain a characterization of the general solutions to Problem 1.

**Corollary A.3** (Solution to Problem 1). *Assume that the matrices $C_1$ and $C_2$ have norm-decreasing columns. Then, for any scaling parameter $\lambda > 0$, it holds that*

$$sens_\lambda^2(C_1, C_2) = \zeta^2 \alpha_1^2 r_d(\lambda \zeta^2 \beta_1^2 / \alpha_1^2) \tag{21}$$

*where $r_d$ is specified in (19) or (20), and $\alpha_1^2$ and $\beta_1^2$ are the squared norms of the first columns of the matrices $C_1$ and $C_2$, respectively.*

*Proof.* A straightforward calculation shows

$$\begin{aligned} sens_\lambda^2(C_1, C_2) &= \max_{i=1,\ldots,n} \sup_{\|x\| \leq \zeta, \|y\| \leq \zeta} \alpha_i^2 \|x - y\|_2^2 + \lambda \beta_i^2 \|x \otimes x - y \otimes y\|_2^2 \\ &= \zeta^2 \alpha_1^2 \left[ \sup_{\|x\| \leq 1, \|y\| \leq 1} \|x - y\|_2^2 + \frac{\lambda \zeta^2 \beta_1^2}{\alpha_1^2} \|x \otimes x - y \otimes y\|_2^2 \right] \\ &= \zeta^2 \alpha_1^2 r_d(\lambda \zeta^2 \beta_1^2 / \alpha_1^2) \end{aligned} \tag{22}$$

completing the proof of Corollary A.3. $\qquad\square$

Corollary A.3 implies that the joint estimate (15) will be $(\epsilon, \delta)$-private, if we use noise of strength at least, $\sigma = \zeta^2 \alpha_1 r_d(\lambda \zeta^2 \beta_1 / \alpha_1) \sigma_{\epsilon,\delta}$, where $\sigma_{\epsilon,\delta}$ is the noise strength required for the Gaussian mechanism with sensitivity 1. This finishes the proof of the Lemma 3.5.

The claim of Theorem 3.2 follows, because Algorithm 1 corresponds exactly to the above construction, only making use of the identity $B(CX + Z) = A(X + C^{-1}Z)$, and for the special case of $\lambda := \lambda^*$, defined as

$$\lambda^* := \frac{\alpha_1^2}{\beta_1^2 \zeta^2 c_d}, \tag{23}$$

with $c_d = 2$ if $d > 1$ and $\frac{8}{11 + 5\sqrt{5}}$ for $d = 1$, which is the smallest values for $\lambda$, such that $r_d\left(\frac{\lambda \zeta^2 \beta_1^2}{\alpha_1^2}\right) = 4$. The sensitivity is then equal to $sens_{\lambda^*}(C_1, C_2) = 2\zeta\alpha_1 = 2\zeta \|C_1\|_{1 \to 2}$, where the last identity holds because of $C_1$'s decreasing column norm structure.

We furthermore note that this value is exactly the sensitivity of estimating the first moment alone, because

$$\max_{X \sim X'} \|C_1 X - C_1 X'\|^2 = \max_{i=1,\ldots,n} \max_{\substack{\|x\| \leq \zeta \\ \|y\| \leq \zeta}} \alpha_i \|x - y\|^2 = 4\alpha_1^2 \zeta^2 \tag{24}$$

This means that JME estimates the second moment without increasing the noise for the first moment, proving our claim that we obtain the **second moment privacy for free**.

## A.2   Proof of Theorem 3.3

To prove Theorem 3.3, we have to determine how the left-hand side of (15) changes in expectation due to the added noise on the right-hand side. Due to the additive nature of the moment estimation process, we can do so explicitly.

For $Z_1 \sim [\mathcal{N}(0, \sigma^2 I)]^d$ it holds that

$$\mathbb{E}_Z \|Y - \hat{Y}\|_F^2 = \mathbb{E}_Z \|B_1 Z_1\|^2 = d\sigma^2 \|B_1\|_F^2. \tag{25}$$

Inserting $B_1 = A_1 C_1^{-1}$ and $\sigma = 2\zeta\sigma_{\epsilon,\delta}\|C_1\|_{1\to 2}$, Equation (3) follows. Analogously, for $Z_2 \sim [\mathcal{N}(0,\sigma^2 I)]^{d\times d}$,

$$\mathbb{E}_Z \|S - \hat{S}\|_{\mathrm{F}}^2 = \mathbb{E}_Z \|\frac{1}{\sqrt{\lambda^*}} B_2 Z_2\|^2 = \frac{d^2\sigma^2}{\lambda^*}\|B_2\|_{\mathrm{F}}^2. \tag{26}$$

With $\sigma$ as above, Equation (4) follows from $B_2 = A_2 C_2^{-1}$, and JME's specific choice of $\lambda^* = \frac{\|C_1\|_{1\to 2}^2}{\|C_2\|_{1\to 2}^2 \zeta^2 c_d}$.

### A.3 Proof of Theorem 4.3

The proof of Theorem 4.3 goes similarly to 3.2, but now we estimate only the diagonal of the second moment matrix. We will show that the sensitivity remains unchanged by proving that the corresponding function $r_d(\lambda)$ in (18) also remains unchanged. Specifically:

$$r_d^{\mathrm{diag}}(\lambda) = \sup_{\|x\|_2 \leq 1, \|y\|_2 \leq 1} \left[\|x - y\|_2^2 + \lambda\|\operatorname{diag}(x \otimes x) - \operatorname{diag}(y \otimes y)\|_2^2\right] = r_d(\lambda) \tag{27}$$

For simplicity, we denote $x \circ x = \operatorname{diag}(x \otimes x)$.

In dimension $d = 1$, the two functions are identical by construction. In dimension $d = 2$, we compute the new function explicitly using the following lemma:

**Lemma A.4.** *Consider $x, y \in \mathbb{R}^2$, and let $\lambda > 0$ then,*

$$r_2^{diag}(\lambda) = \sup_{\|x\|_2 \leq 1, \|y\|_2 \leq 1} \left[\|x - y\|_2^2 + \lambda\|x \circ x - y \circ y\|_2^2\right] = \begin{cases} 4, & \text{if } \lambda \leq \frac{1}{2}, \\ 2 + 2\lambda + \frac{1}{2\lambda}, & \text{if } \lambda > \frac{1}{2}. \end{cases} \tag{28}$$

Next, we use a dimension reduction argument to prove that $r_d^{\mathrm{diag}}(\lambda) = r_2^{\mathrm{diag}}(\lambda)$ for all $d \geq 2$, via the following lemma:

**Lemma A.5** (Dimension Reduction). *For any vectors $x, y \in \mathbb{R}^d$, where $d \geq 3$, there exist vectors $x', y' \in \mathbb{R}^{d-1}$ that for any $\lambda > 0$ satisfies the inequality:*

$$\|x - y\|_2^2 + \lambda\|x \circ x - y \circ y\|_2^2 \leq \|x' - y'\|_2^2 + \lambda\|x' \circ x' - y' \circ y'\|_2^2. \tag{29}$$

We apply this lemma recursively to prove that $r_d^{\mathrm{diag}}(\lambda) = r_2^{\mathrm{diag}}(\lambda)$ for all $d \geq 2$. The proof of the lemma can be found later in the appendix.

By combining these lemmas, we conclude the proof of the theorem.

# B Additional materials

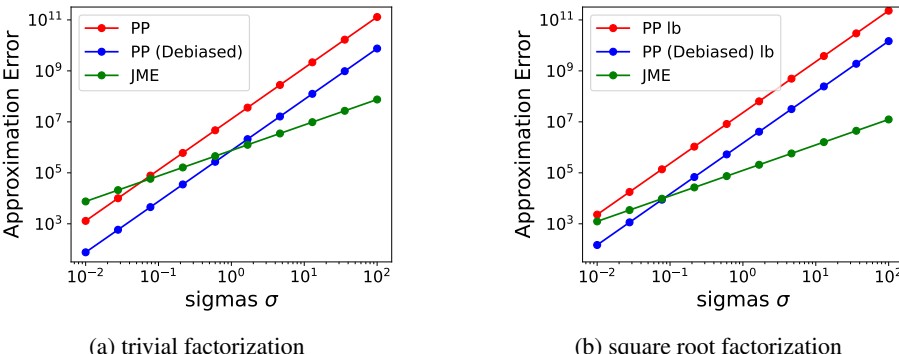

(a) trivial factorization  (b) square root factorization

Figure 5: Expected error of the estimated covariance vector for Adam with JME versus PP ($d = 10^6$; $n = 1000$). JME provides a better estimate in the mid to high privacy regime.

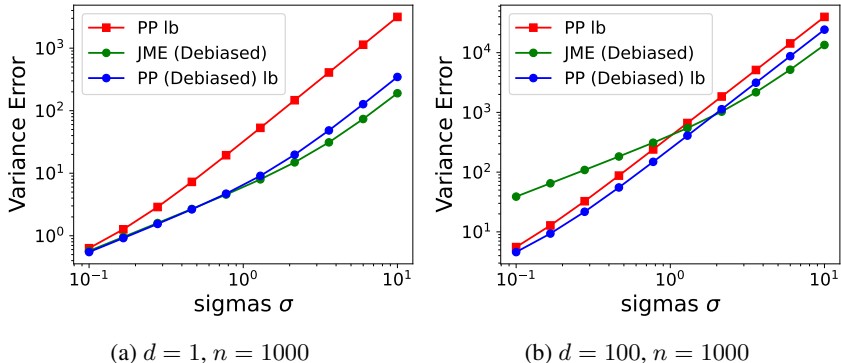

(a) $d = 1$, $n = 1000$  (b) $d = 100$, $n = 1000$

Figure 6: Expected error of the estimated *covariance matrix* with JME versus PP (with trivial factorizations). For $d = 1$, JME consistently achieves quality better than or on par with PP. For $d = 100$, JME is preferable to PP only in the high privacy regime.

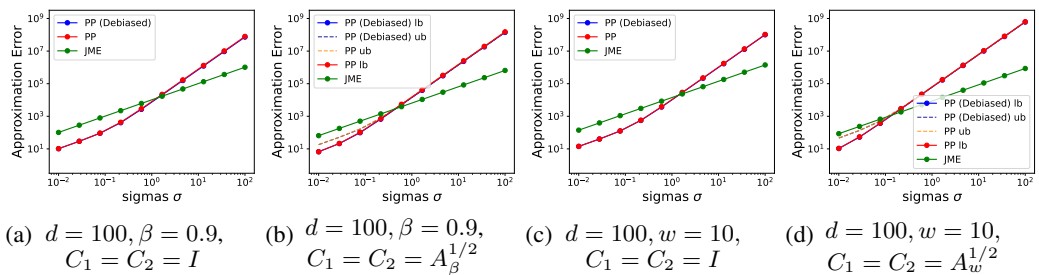

(a) $d = 100, \beta = 0.9$, $C_1 = C_2 = I$  (b) $d = 100, \beta = 0.9$, $C_1 = C_2 = A_\beta^{1/2}$  (c) $d = 100, w = 10$, $C_1 = C_2 = I$  (d) $d = 100, w = 10$, $C_1 = C_2 = A_w^{1/2}$

Figure 7: Expected error of second moment estimation with JME versus PP under different workload settings. The scenarios include exponential decay ($\beta = 0.9$) and sliding window ($w = 10$) workloads and $d = 100, n = 1000$, both trivial and square root matrix factorization. In line with our analysis, incorporating matrix factorization significantly improves the quality of both methods, particularly in the high privacy regime.

Table 2: CIFAR-10 experiments with two different privacy budgets, $\varepsilon \approx 1.7$ and $\varepsilon \approx 0.16$ for $\delta = 10^{-6}$, for four methods: DP-Adam with and without debiasing, JME, and Joint Clipping. The average and standard deviation errors are based on 3 runs.

| | Method | Epoch 1 | Epoch 2 | Epoch 3 | Epoch 4 | Epoch 5 | Epoch 6 | Epoch 7 | Epoch 8 | Epoch 9 | Epoch 10 |
|---|---|---|---|---|---|---|---|---|---|---|---|
| $\varepsilon \approx 0.16$ | DP-Adam-JME | $12.43 \pm 3.95$ | $19.18 \pm 2.27$ | $23.29 \pm 1.25$ | $25.83 \pm 0.48$ | $27.25 \pm 0.20$ | $27.81 \pm 0.14$ | $28.50 \pm 0.40$ | $28.89 \pm 0.54$ | $29.61 \pm 0.51$ | $30.38 \pm 0.62$ |
| | DP-Adam-Clip | $13.54 \pm 3.31$ | $19.68 \pm 2.76$ | $24.92 \pm 1.48$ | $26.24 \pm 1.04$ | $26.60 \pm 0.25$ | $27.27 \pm 0.55$ | $27.56 \pm 0.46$ | $27.98 \pm 0.26$ | $27.78 \pm 0.17$ | $28.14 \pm 0.33$ |
| | DP-Adam-Debiased | $12.41 \pm 4.14$ | $16.91 \pm 1.36$ | $23.78 \pm 0.70$ | $24.92 \pm 0.55$ | $26.20 \pm 0.65$ | $26.96 \pm 1.19$ | $27.46 \pm 0.90$ | $28.14 \pm 0.72$ | $28.55 \pm 0.92$ | $29.14 \pm 0.60$ |
| | DP-Adam | $10.29 \pm 0.51$ | $10.32 \pm 0.56$ | $10.32 \pm 0.56$ | $10.34 \pm 0.58$ | $10.39 \pm 0.67$ | $10.41 \pm 0.70$ | $10.42 \pm 0.73$ | $10.45 \pm 0.79$ | $10.45 \pm 0.78$ | $10.48 \pm 0.83$ |
| $\varepsilon \approx 1.7$ | DP-Adam-JME | $32.64 \pm 0.71$ | $39.19 \pm 1.52$ | $42.28 \pm 0.48$ | $44.44 \pm 0.57$ | $45.12 \pm 0.81$ | $45.61 \pm 0.73$ | $46.91 \pm 1.21$ | $47.04 \pm 0.98$ | $47.24 \pm 0.79$ | $47.94 \pm 0.85$ |
| | DP-Adam-Clip | $30.51 \pm 0.72$ | $38.03 \pm 1.00$ | $40.66 \pm 0.75$ | $42.88 \pm 0.46$ | $43.30 \pm 0.71$ | $43.52 \pm 0.50$ | $44.51 \pm 0.70$ | $44.45 \pm 0.24$ | $44.76 \pm 0.22$ | $44.34 \pm 0.28$ |
| | DP-Adam-Debiased | $34.21 \pm 0.95$ | $41.38 \pm 1.54$ | $45.20 \pm 0.44$ | $46.91 \pm 1.25$ | $48.06 \pm 0.36$ | $48.38 \pm 0.83$ | $48.26 \pm 1.05$ | $48.38 \pm 1.21$ | $47.81 \pm 0.73$ | $48.53 \pm 0.99$ |
| | DP-Adam | $26.58 \pm 0.43$ | $31.45 \pm 0.43$ | $35.02 \pm 1.16$ | $36.39 \pm 0.58$ | $38.15 \pm 0.95$ | $39.31 \pm 0.52$ | $40.33 \pm 0.33$ | $41.47 \pm 1.05$ | $42.42 \pm 0.94$ | $43.07 \pm 0.86$ |

Table 3: Hyperparameters for CIFAR-10 Experiments. Medium-privacy experiments use a batch size of 256, compared to 1 in the high-privacy regime, while using a noise multiplier of $\sigma_{\varepsilon,\delta} = 1$ for the medium-privacy regime and $\sigma_{\varepsilon,\delta} = 2$ for the high-privacy regime. JME and joint clipping require an additional hyperparameter—scaling—which is optimized to find the best value for those runs. We also find it helpful to clip the updates; for this, we use the same clipping norm.

| | Noise Mult | Batch Size | Method | lr | Scaling $(\lambda, \tau)$ | eps | Clip Norm |
|---|---|---|---|---|---|---|---|
| $\varepsilon \approx 1.7$ | 1 | 256 | DP-Adam-JME | $10^{-3}$ | 1 | $10^{-6}$ | 1 |
| | | | DP-Adam-Clip | $10^{-3}$ | 0.5 | $10^{-6}$ | 1 |
| | | | DP-Adam-Debiased | $10^{-3}$ | - | $10^{-6}$ | 1 |
| | | | DP-Adam | $10^{-3}$ | - | $10^{-8}$ | 1 |
| $\varepsilon \approx 0.16$ | 2 | 1 | DP-Adam-JME | $10^{-7}$ | 1 | $10^{-7}$ | 1 |
| | | | DP-Adam-Clip | $10^{-7}$ | 0.5 | $10^{-7}$ | 1 |
| | | | DP-Adam-Debiased | $10^{-7}$ | - | $10^{-7}$ | 1 |
| | | | DP-Adam | $10^{-7}$ | - | $10^{-8}$ | 1 |

# C Algorithms

---

**Algorithm 2** Differentially Private JME Adam

---

**Input:** Initial model $\theta_0 \in \mathbb{R}^d$, dataset $D$, batchsize $b$, matrices $C_{\beta_1}, C_{\beta_2} \in \mathbb{R}^{n \times n}$, model loss $\ell(\theta, d)$, clipnorm $\zeta$, noise multiplier $\sigma_{\epsilon, \delta} \geq 0$, learning rate $\alpha > 0$, and parameters $\beta_1 = 0.9$, $\beta_2 = 0.999$, $\varepsilon = 10^{-8}$.

$m_0 \leftarrow 0$     // first moment initialization.

$v_0 \leftarrow 0$     // second moment initialization.

$\lambda, s_\lambda = \cdot Joint\text{-}sens(C_{\beta_1}, C_{\beta_2})$     // joint sensitivity

$Z_1, Z_2 \sim N(0, \sigma_{\epsilon,\delta}^2 s_\lambda^2 I_d)$     // noise generating

**for** $i = 1, 2, \ldots, n$ **do**

    $S_i \leftarrow \{d_1, \ldots, d_m\} \subseteq D$    select a data batch

    $g_j \leftarrow \nabla_\theta \ell(\theta_{i-1}, d_j))$    for $j = 1, \ldots, m$

    $x_i \leftarrow \sum_{j=1}^m \min(1, \zeta/\|g_j\|) g_j$

    $\widehat{x}_i \leftarrow x_i + \zeta [C_{\beta_1}^{-1} Z_1]_{[i, \cdot]}$

    $\widehat{x^2}_i \leftarrow x_i^2 + \lambda^{-1/2} \zeta [C_{\beta_2}^{-1} Z_2]_{[i, \cdot]}$

    $m_i \leftarrow m_{i-1} \beta_1 + (1 - \beta_1) \widehat{x}_i$

    $v_i \leftarrow v_{i-1} \beta_2 + (1 - \beta_2) \widehat{x^2}_i$

    $\widehat{m}_i = m_i / (1 - \beta_1^i)$     // bias-correction

    $\widehat{v}_i = v_i / (1 - \beta_2^i)$     // bias-correction

    $\theta_i \leftarrow \theta_{i-1} - \alpha \widehat{m}_i / (\sqrt{\widehat{v}_i} + \epsilon)$

**end for**

**Ensure:** $\Theta = (\theta_1, \ldots, \theta_n)$

---

---

**Algorithm 3** $\lambda$-JME

---

**Require:** input stream vectors $x_1, \ldots, x_n \in \mathbb{R}^d$ with $\|x_t\|_2 \leq \zeta$ for $\zeta > 0$

**Require:** workload matrices $A_1 = (a_k^t), A_2 = (b_k^t) \in \mathbb{R}^{n \times n}$

**Require:** noise shaping matrices $C_1, C_2$ (lower triangular, invertible, decreasing column norms) (default: $I_{n \times n}$)

**Require:** privacy parameters $(\epsilon, \delta)$

    $\sigma_{\epsilon, \delta} \leftarrow$ noise strength for $(\epsilon, \delta)$-dp Gaussian mechanism

    $s \leftarrow \zeta \|C_1\|_{1 \to 2} r_d \left( \frac{\lambda \zeta^2 \|C_2\|_{1 \to 2}^2}{\|C_1\|_{1 \to 2}^2} \right)^{1/2}$     // joint sensitivity

    $Z_1 \sim \left[ \mathcal{N}(0, \sigma_{\epsilon,\delta}^2 s^2) \right]^{n \times d}$     // 1st moment noise

    $Z_2 \sim \left[ \mathcal{N}(0, \sigma_{\epsilon,\delta}^2 s^2) \right]^{n \times d \times d}$     // 2nd moment noise

    **for** $t = 1, 2, \ldots, n$ **do**

        $\widehat{x}_t \leftarrow x_t + [C_1^{-1} Z_1]_{[t, \cdot]}$

        $\widehat{x_t \otimes x_t} \leftarrow x_t \otimes x_t + \lambda^{-1/2} [C_2^{-1} Z_2]_{[t, \cdot, \cdot]}$

        **yield**   $\widehat{Y}_t = \sum_{k=1}^t a_k^t \widehat{x_k}, \quad \widehat{S}_t = \sum_{k=1}^t b_k^t \widehat{x_k \otimes x_k}$

    **end for**

---

---

**Algorithm 4** $\alpha$-IME (IME with budget split parameter $\alpha \in (0,1)$)

---

**Require:** input stream of vectors $x_1, \ldots, x_n \in \mathbb{R}^d$ with $\|x_t\|_2 \leq \zeta$ for $\zeta > 0$
**Require:** workload matrices $A_1 = (a_k^t)$, $A_2 = (b_k^t) \in \mathbb{R}^{n \times n}$
**Require:** noise shaping matrices $C_1, C_2$ (lower triangular, invertible, decreasing column norms)
  (default: $\mathrm{I}_{n \times n}$)
**Require:** privacy parameters $(\epsilon, \delta)$
**Require:** $\sigma \leftarrow$ noise strength for $(\epsilon, \delta)$-dp Gaussian mechanism
**Require:** privacy trade-off $\alpha \in (0,1)$

$\quad \sigma_1 \leftarrow \frac{\sigma}{\sqrt{\alpha}}$
$\quad s_1 \leftarrow 2\zeta \|C_1\|_{1 \to 2}$                       // sensitivity of 1st moment
$\quad Z_1 \sim \left[\mathcal{N}(0, \sigma_1^2 s_1^2)\right]^{n \times d}$                   // 1st moment noise
$\quad \sigma_2 \leftarrow \frac{\sigma}{\sqrt{1-\alpha}}$
$\quad s_2 \leftarrow \sqrt{2}\zeta^2 \|C_2\|_{1 \to 2}$                    // sensitivity of 2nd moment
$\quad Z_2 \sim \left[\mathcal{N}(0, \sigma^2 s_2^2)\right]^{n \times d \times d}$               // 2nd moment noise
$\quad$**for** $t = 1, 2, \ldots, n$ **do**
$\quad\quad \widehat{x_t} \leftarrow x_t + [C_1^{-1} Z_1]_{[t, \cdot]}$
$\quad\quad \widehat{x_t \otimes x_t} \leftarrow x_t \otimes x_t + [C_2^{-1} Z_2]_{[t, \cdot, \cdot]}$
$\quad\quad$**yield** $\quad \widehat{Y}_t = \sum\limits_{k=1}^{t} a_k^t \widehat{x_k}, \quad \widehat{S}_t = \sum\limits_{k=1}^{t} b_k^t \widehat{x_k \otimes x_k}$
$\quad$**end for**

---




---

**Algorithm 5** $\tau$-CS (CS with 2nd moment rescaling parameter $\tau > 0$)

---

**Require:** input stream of vectors $x_1, \ldots, x_n \in \mathbb{R}^d$ with
**Require:** workload matrices $A = (a_k^t) \in \mathbb{R}^{n \times n}$
**Require:** input dimension $d$, bound on input norm $\zeta > 0$
**Require:** privacy parameters $(\epsilon, \delta)$
**Require:** (optional) noise shaping matrix $C$ (lower triangular, invertible, decreasing column norms)
  (default: $\mathrm{I}_{n \times n}$)

$\quad \sigma_{\epsilon, \delta} \leftarrow$ noise strength for $(\epsilon, \delta)$-dp Gaussian mechanism
$\quad s \leftarrow 2\zeta\sqrt{1 + \tau\zeta^2}$             // sensitivity based on norm of concatenated data
$\quad Z \sim \left[\mathcal{N}(0, \sigma_{\epsilon, \delta}^2 s^2)\right]^{n \times d(d+1)}$        // noise matrix for concatenated data
$\quad$**for** $t = 1, 2, \ldots, n$ **do**
$\quad\quad \widetilde{x}_t = \left( x_t, \sqrt{\tau}\mathsf{vec}(x_t \otimes x_t) \right)$
$\quad\quad \widehat{\widetilde{x}}_t \leftarrow \widetilde{x}_t + [C^{-1} Z]_{[t, \cdot]}$
$\quad\quad$**yield** $\quad \widehat{Y}_t = \sum\limits_{k=1}^{t} a_k^t [\widehat{\widetilde{x}}_t]_{1:d} \quad \widehat{S}_t = \frac{1}{\sqrt{\tau}} \sum\limits_{k=1}^{t} b_k^t [\widehat{\widetilde{x}}_t]_{(d+1):d(d+1)}$
$\quad$**end for**

---

**Algorithm 6** PP

---

**Require:** input stream of vectors $x_1, \ldots, x_n \in \mathbb{R}^d$ with $\|x_t\|_2 \leq \zeta$ for $\zeta > 0$
**Require:** workload matrices $A_1 = (a_k^t), A_2 = (b_k^t) \in \mathbb{R}^{n \times n}$
**Require:** noise shaping matrix $C_1$ (lower triangular, invertible, decreasing column norms)    (default: $I_{n \times n}$)
**Require:** privacy parameters $(\epsilon, \delta)$

   $\sigma_{\epsilon,\delta} \leftarrow$ noise strength for $(\epsilon, \delta)$-dp Gaussian mechanism
   $s \leftarrow 2\zeta\|C_1\|_{1 \rightarrow 2}$                            // sensitivity of 1st moment
   $Z \sim \left[\mathcal{N}(0, \sigma_{\epsilon,\delta}^2 s^2)\right]^{n \times d}$                       // 1st moment noise
   **for** $t = 1, 2, \ldots, n$ **do**
      $\widehat{x}_t \leftarrow x_t + [C_1^{-1} Z]_{[t, \cdot]}$
      $b_t \leftarrow I_{d \times d} \times \sigma_{\epsilon,\delta}^2 \|C_1\|_{1 \rightarrow 2}^2 \sum_{k=1}^{n} (A_2)_{t,k} (C_1 C_1^\top)_{k,k}^{-1}$       // bias term (optional)
      $\widehat{x_t \otimes x_t} \leftarrow \widehat{x}_t \otimes \widehat{x}_t - b_t$
      **yield**   $\widehat{Y}_t = \sum_{k=1}^{t} a_k^t \widehat{x}_k, \quad \widehat{S}_t = \sum_{k=1}^{t} b_k^t \widehat{x_k \otimes x_k}$
   **end for**

---

# D  Technical Proofs

**Definition D.1** (Face-Splitting Product). Let $A = (a_i^\top)_{i=1}^n$ with $a_i \in \mathbb{R}^{d_1}$ and $B = (b_i^\top)_{i=1}^n$ with $b_i \in \mathbb{R}^{d_2}$. The *Face-Splitting Product* of $A$ and $B$, denoted by $A \bullet B$, is defined as:

$$A \bullet B = (a_i \otimes b_i)_{i=1}^n = (\text{vec}(a_i b_i^\top))_{i=1}^n, \tag{30}$$

where $\otimes$ denotes the Kronecker product, and $\text{vec}$ denotes vectorization.

The result is a matrix of size $n \times (d_1 d_2)$, where each row corresponds to the vectorized Kronecker product of $a_i$ and $b_i$. A few properties of the Face-Splitting Product that we will use further:

- *Bilinearity $A \bullet (B + C) = A \bullet B + A \bullet C$ and $(A + B) \bullet C = A \bullet C + B \bullet C$).*
- *Associativity $(A \bullet B) \bullet C = A \bullet (B \bullet C)$.*
- *Frobenius norm $\|A \bullet B\|_F = \|B \bullet A\|_F$, even though $A \bullet B \neq B \bullet A$.*

## D.1  Bounds on the expected approximation error for PP with non-trivial factorization

**Lemma D.2.**

$$\|A_2 C_1^{-1}\|_F^2 \leq \sup_{X \in \mathcal{X}} \text{tr}\left((A_2^\top A_2 \circ C_1^{-1} C_1^{-\top}) X X^\top\right) \leq \sum_{ij} \left|[A_2^\top A_2 \circ C_1^{-1} C_1^{-\top}]_{i,j}\right| \tag{31}$$

*Proof of Lemma D.2.* The proof is elementary. For the lower bound. consider the specific choice for $X = (x_1, \ldots, x_n)$ with all rows identical with $\|x_i\| = 1$, such that $XX^\top$ is the constant matrix of all 1s. For the upper bound, we observe that $[XX^\top]_{i,j} \in [-1, 1]$, so

$$\text{tr}\left((A_2^\top A_2 \circ C_1^{-1} C_1^{-\top}) X X^\top\right) = \sum_{i,j}[A_2^\top A_2 \circ C_1^{-1} C_1^{-\top}]_{i,j}[XX^\top]_{i,j} \leq \sum_{i,j}\left|[A_2^\top A_2 \circ C_1^{-1} C_1^{-\top}]_{i,j}\right|. \tag{32}$$

$\square$

**Theorem A.1** ($d = 1$). *For any $\lambda > 0$, it holds:*

$$r_1(\lambda) = \begin{cases} 4 & \text{if } \lambda \leq \frac{11 + 5\sqrt{5}}{8}, \\ \frac{1}{8}(3 - \tau)^2(\lambda\tau + 1 + \lambda) & \text{otherwise.} \end{cases} \tag{19}$$

*where $\tau = \sqrt{1 - 2/\lambda}$. Moreover, the function $r_1(\lambda)$ is a continuous function with respect to the parameter $\lambda > 0$.*

*Proof.* We recall that the function $r_1(\lambda)$ is defined as

$$r_1(\lambda) = \sup_{|x|, |y| \leq 1} [(x - y)^2 + \lambda(x^2 - y^2)^2]. \tag{33}$$

Given the difference it is an increasing function of $x + y$ therefore without any loss of generality we can assume $y = 1$. Then the derivative of this expression would give us

$$2(x - 1)(1 + 2\lambda(x^2 + x)) \tag{34}$$

The optimal value will depend on the roots of the quadratic polynomial. The determinant is $4\lambda^2 - 8\lambda$ therefore for $\lambda < 2$ there are no roots, the function is decreasing on the segment $[-1, 1]$ reaching its maximal value at $x = -1$ with the value 4. Otherwise we have the following roots $\frac{-1 \pm \sqrt{1 - 2/\lambda}}{2}$. We observe that for large $\lambda \gg 1$ we have the optimal $x = 0$ which corresponds to the maximal value for the second term. The maximum could be in both $x = -1$ and $x = \frac{-1 + \sqrt{1 - 2/\lambda}}{2}$ therefore we will need to compare them. By substituting it into the expression we get $\frac{1}{8}(\tau - 3)^2(\lambda\tau + 1 + \lambda)$, where $\tau = \sqrt{1 - 2/\lambda}$. We should compare this expression with 4 to find a boundary when $x = -1$ is an optimal solution.

$$\frac{1}{8}(\tau - 3)^2(\lambda\tau + 1 + \lambda) \le 4 \tag{35}$$

We can express $1/\lambda = \frac{1-\tau^2}{2}$. Therefore we get the following inequality of variable $0 \le \tau < 1$:

$$\begin{aligned}
\frac{1}{8}(\tau - 3)^2(\tau + \frac{1-\tau^2}{2} + 1) - 4\frac{1-\tau^2}{2} &= \frac{1}{16}(3-\tau)^3(1+\tau) - 2(1-\tau)(1+\tau) \\
&= \frac{1}{16}(1+\tau)\left[27 - 27\tau + 9\tau^2 - \tau^3 - 32 + 32\tau\right] \quad (36) \\
&= -\frac{1}{16}(1+\tau)^2(\tau^2 - 10\tau + 5) \le 0
\end{aligned}$$

Which is less than 0 for $\tau \le 5 - 2\sqrt{5} \Rightarrow \lambda \le \frac{11+5\sqrt{5}}{8} \approx 2.77$ which concludes the proof. $\qquad\square$

**Theorem A.2** (Joint Sensitivity for Moments Estimation). *For $d > 1$ and $\lambda > 0$:*

$$r_d(\lambda) = \begin{cases} 4 & \text{if } \lambda \le \frac{1}{2}, \\ 2 + 2\lambda + \frac{1}{2\lambda} & \text{otherwise.} \end{cases} \tag{20}$$

*Proof.* First, we decompose the Kronecker product as follows:

$$\begin{aligned}
\|x \otimes x - y \otimes y\|_F^2 = \|xx^T - yy^T\|_F^2 &= \operatorname{tr}(xx^T xx^T) - 2\operatorname{tr}(xx^T yy^T) + \operatorname{tr}(yy^T yy^T) \\
&= \|x\|_2^4 + \|y\|_2^4 - 2\langle x, y\rangle^2.
\end{aligned} \tag{37}$$

Similarly, the squared norm of the difference is

$$\|x - y\|_2^2 = \|x\|_2^2 + \|y\|_2^2 - 2\langle x, y\rangle. \tag{38}$$

Therefore, the objective function becomes

$$\|x - y\|^2 + \lambda\|x \otimes x - y \otimes y\|_F^2 = \|x\|_2^2 + \|y\|_2^2 + \lambda\|x\|_2^4 + \lambda\|y\|_2^4 - 2\langle x, y\rangle - 2\lambda\langle x, y\rangle^2. \tag{39}$$

The first four terms are maximized when $\|x\| = \|y\| = 1$, yielding $2 + 2\lambda$. The last two terms can then be optimized independently over the scalar product $\langle x, y\rangle$ in dimension $d > 1$.

Let $\beta = \langle x, y\rangle$, where $-1 \le \beta \le 1$. The expression $-2\beta - 2\lambda\beta^2$ is maximized at $\beta = -\frac{1}{2\lambda}$ when $\lambda \ge \frac{1}{2}$, or at $\beta = -1$ when $\lambda < \frac{1}{2}$. If $\beta = -1$, then $x = -y$ and the objective function becomes 4. If $\beta = -\frac{1}{2\lambda}$, the objective function is equal to $2 + 2\lambda + \frac{1}{2\lambda}$, which concludes the proof.

$$\square$$

**Theorem 3.6** (JME vs IME). *For any $\epsilon, \delta > 0$, $\lambda$-JME Pareto-dominates IME with respect to the approximation error for the first vs second moment estimates.*

*Proof.* We begin the proof with the following observation: $\lambda$-JME and IME introduce *additive* noise to the first and second moments. Therefore, for the Frobenius approximation error, it is sufficient to compare just the variances of the noise introduced by those methods. We use an instance of a Gaussian mechanism, and the privacy guarantees are more appropriately characterized by the notion of Gaussian privacy. For the sake of the proof, we assume that $(\epsilon, \delta)$-DP is equivalent to $\mu$-GDP for a specific choice of $\mu$. The sensitivity of $\lambda$-JME depends on the dimensionality. Here, we assume $d > 1$ to address the hardest case, as $d = 1$ follows trivially.

To estimate $x$ and $x \otimes x$ simultaneously in a differentially private way via composition theorem, we need to split the privacy budget between the components. Using the Gaussian mechanism, we split the budget as $\mu_1$-GDP for the first component and $\mu_2$-GDP for the second component such that $\mu_1^2 + \mu_2^2 = \mu^2$. The squared sensitivity of $x$ is $4\zeta^2$, and the squared sensitivity of $x \otimes x$ is $2\zeta^4$,

assuming $\|x\|_2 \leq \zeta$. Therefore, the variance of noise added to the first and second components is given by:

$$\left( \frac{4\zeta^2}{\mu_1^2}, \frac{2\zeta^4}{\mu^2 - \mu_1^2} \right). \tag{40}$$

Our analysis for JME yields the pair of variances, for $\lambda\zeta^2 \geq \frac{1}{2}$:

$$\left( \frac{\zeta^2(2 + 2\lambda\zeta^2 + \frac{1}{2\lambda\zeta^2})}{\mu^2}, \frac{\zeta^2(2 + 2\lambda\zeta^2 + \frac{1}{2\lambda\zeta^2})}{\lambda\mu^2} \right). \tag{41}$$

We aim to show that for a given variance in the first component, our variance for the second component is smaller. Specifically, we need to prove:

$$\frac{\zeta^2(2 + 2\lambda\zeta^2 + \frac{1}{2\lambda\zeta^2})}{\lambda\mu^2} < \frac{2\zeta^4}{\mu^2 - \mu_1^2}, \quad \text{where} \quad \frac{4\zeta^2}{\mu_1^2} = \frac{\zeta^2(2 + 2\lambda\zeta^2 + \frac{1}{2\lambda\zeta^2})}{\mu^2}. \tag{42}$$

By substituting $\frac{\mu^2}{\mu_1^2} = \frac{1}{2} + \frac{\lambda\zeta^2}{2} + \frac{1}{8\lambda\zeta^2}$ into the inequality, we obtain:

$$\frac{2\lambda\zeta^2}{2 + 2\lambda\zeta^2 + \frac{1}{2\lambda\zeta^2}} > 1 - \frac{1}{\frac{1}{2} + \frac{\lambda\zeta^2}{2} + \frac{1}{8\lambda\zeta^2}}. \tag{43}$$

Multiplying both sides by the denominator yields:

$$2\lambda\zeta^2 > 2 + 2\lambda\zeta^2 + \frac{1}{2\lambda\zeta^2} - 4. \tag{44}$$

Simplifying, we find $\lambda\zeta^2 > \frac{1}{4}$, which is satisfied by the initial assumption. $\square$

**Theorem 3.7** (JME vs CS). *For any $\epsilon, \delta > 0$, $\lambda$-JME Pareto-dominates CS with respect to the approximation error for the first vs second moment estimates.*

*Proof.* We begin the proof with the following observation: $\lambda$-JME and CS introduce *additive* noise to the first and second moments. Therefore, for the Frobenius approximation error, it is sufficient to compare just the variances of the noise introduced by those methods. We use an instance of a Gaussian mechanism, and the privacy guarantees are more appropriately characterized by the notion of Gaussian privacy. For the sake of the proof, we assume that $(\epsilon, \delta)$-DP is equivalent to $\mu$-GDP for a specific choice of $\mu$. The sensitivity of $\lambda$-JME depends on the dimensionality. Here, we assume $d > 1$ to address the hardest case, as $d = 1$ follows trivially.

We aim to show that Joint Moment Estimation (JME) introduces less noise to the $(x \otimes x)$ component than CS under the same privacy budget $\mu$-GDP. Specifically, we compare the variances in the second coordinate under the assumption that the noise variances in the first coordinate are equal.

The combined vector of $x_i$ and $x_i \otimes x_i$ can be represented as:

$$(x_i, \sqrt{\tau} x_i \otimes x_i), \tag{45}$$

where $\tau > 0$ is a scaling parameter. If the input vectors $x_i$ are bounded by $\|x_i\|_2 \leq \zeta$, the resulting vector will have $l_2$ norm bounded by $\sqrt{\zeta^2 + \tau\zeta^4}$. The squared sensitivity of the vector is therefore: $4\zeta^2(1 + \tau\zeta^2)$. This results in the following variances for the first and second coordinates under noise addition:

$$\left( \frac{4\zeta^2(1 + \tau\zeta^2)}{\mu^2}, \frac{4\zeta^2(1 + \tau\zeta^2)}{\tau\mu^2} \right). \tag{46}$$

For JME, we analyze the variances and obtain the pair of variances as:

$$\left( \frac{\zeta^2(2 + 2\zeta^2\lambda + \frac{1}{2\zeta^2\lambda})}{\mu^2}, \frac{\zeta^2(2 + 2\zeta^2\lambda + \frac{1}{2\zeta^2\lambda})}{\lambda\mu^2} \right), \tag{47}$$

where we assume $\zeta^2\lambda \geq \frac{1}{2}$.

To compare the noise introduced to the second component, we aim to show:

$$\frac{\zeta^2(2 + 2\zeta^2\lambda + \frac{1}{2\zeta^2\lambda})}{\lambda\mu^2} < \frac{4\zeta^2(1 + \tau\zeta^2)}{\tau\mu^2}, \tag{48}$$

under the condition that the variances in the first component are equal:

$$\frac{\zeta^2(2 + 2\zeta^2\lambda + \frac{1}{2\zeta^2\lambda})}{\mu^2} = \frac{4\zeta^2(1 + \tau\zeta^2)}{\mu^2}. \tag{49}$$

Given the equality, inequality is equivalent to $\lambda > \tau$. Simplifying the equality in the first component gives:

$$2 + 2\zeta^2\lambda + \frac{1}{2\zeta^2\lambda} = 4 + 4\tau\zeta^2. \tag{50}$$

Rearranging terms, we isolate $\frac{1}{2\zeta^2\lambda}$:

$$\frac{1}{2\zeta^2\lambda} = 2 + 2(2\tau - \lambda)\zeta^2. \tag{51}$$

From the assumption $\zeta^2\lambda \geq \frac{1}{2}$, we know that $\frac{1}{2\zeta^2\lambda} < 1$. For this inequality to hold, the term $2 + 2(2\tau - \lambda)\zeta^2$ must also be less than 1. Therefore, $2\tau - \lambda < 0$, therefore $\lambda > \tau$.

Thus, we conclude that for the same variance in the first component, JME introduces less noise variance to the second component compared to CS. $\qquad\square$

**Lemma 3.8** (Expected Second Moment Error for PP). *Assume the same setting as for Theorem 3.3, except that we compute the estimates $\widehat{Y}$ and $\widehat{S}_{PP}$ with the PP method. Let $Q = C_1^{-1}C_1^{-\top}$ and $E_Q = \mathrm{diag}(Q)\mathrm{diag}(Q)^\top$. Then, the expected approximation error of the second moment satisfies:*

$$\sup_{X \in \mathcal{X}} \mathbb{E}\|S - \widehat{S}_{PP}\|_F^2 = 16d(d+1)\sigma_{\epsilon,\delta}^4\zeta^4\|C_1\|_{1\to2}^4 \mathrm{tr}(A_2^\top A_2(Q \circ Q)) \tag{9}$$
$$+ 8(d+1)\sigma_{\epsilon,\delta}^2\zeta^2\|C_1\|_{1\to2}^2 \sup_{X \in \mathcal{X}} \mathrm{tr}((A_2^\top A_2 \circ Q)XX^T) \overbrace{+ 16d\sigma_{\epsilon,\delta}^4\zeta^4\|C_1\|_{1\to2}^4 \mathrm{tr}(A_2^\top A_2 E_Q)}^{bias}.$$

*For debiased PP, the term marked "bias" does not occur.*

*Proof.* We aim to evaluate the expected squared Frobenius norm of the error:

$$\sup_{X \in \mathcal{X}} \mathbb{E}\|A_2((X + C_1^{-1}Z_1)\bullet(X + C_1^{-1}Z_1)) - A_2(X \bullet X)\|_F^2$$
$$= 2\underbrace{\sup_{X \in \mathcal{X}} \mathbb{E}\|A_2(X \bullet C_1^{-1}Z_1)\|_F^2}_{\mathcal{S}_1} + 2\underbrace{\sup_{X \in \mathcal{X}} \mathbb{E}\langle A_2(X \bullet C_1^{-1}Z_1), A_2(C_1^{-1}Z_1 \bullet X)\rangle_F}_{\mathcal{S}_2} \tag{52}$$
$$+ \underbrace{\mathbb{E}\|A_2((C_1^{-1}Z_1)\bullet(C_1^{-1}Z_1))\|_F^2}_{\mathcal{S}_3}.$$

We compute those terms separately. We add component-wise independent Gaussian noise $Z_1 \sim \mathcal{N}(0, \sigma^2)$ to the first moment, where $\sigma = 2\zeta\sigma_{\epsilon,\delta}$.

**Step 1. Bound $\mathcal{S}_1$.**

$$
\begin{aligned}
\mathcal{S}_1 &= \sup_{X \in \mathcal{X}} \mathbb{E} \| A_2(X \bullet C_1^{-1} Z_1) \|_{\mathrm{F}}^2 \\
&= \sup_{X \in \mathcal{X}} \mathbb{E} \sum_{k=1}^{n} \sum_{i,j}^{d} \left( \sum_{t=1}^{n} (A_2)_{k,t} X_{t,i} \sum_{r=1}^{n} (C_1^{-1})_{t,r} (Z_1)_{r,j} \right)^2 \\
&= \sup_{X \in \mathcal{X}} \mathbb{E} \sum_{k=1}^{n} \sum_{i,j}^{d} \sum_{t_1,t_2}^{n} (A_2)_{k,t_1} (A_2)_{k,t_1} X_{t_1,i} X_{t_2,i} \sum_{r=1}^{n} (C_1^{-1})_{t_1,r} (C_1^{-1})_{t_2,r} \mathbb{E}(Z_1)_{r,j}^2 \\
&= \sigma^2 \| C_1 \|_{1\to 2}^2 \sup_{X \in \mathcal{X}} \sum_{k=1}^{n} \sum_{i,j}^{d} \sum_{t_1,t_2}^{n} (A_2)_{k,t_1} (A_2)_{k,t_2} X_{t_1,i} X_{t_2,i} \sum_{r=1}^{n} (C_1^{-1})_{t_1,r} (C_1^{-1})_{t_2,r} \\
&= d\sigma^2 \| C_1 \|_{1\to 2}^2 \sup_{X \in \mathcal{X}} \sum_{t_1,t_2}^{n} \langle (A_2^\top)_{t_1}, (A_2^\top)_{t_2} \rangle \langle X_{t_1}, X_{t_2} \rangle \langle (C_1^{-1})_{t_1}, (C_1^{-1})_{t_2} \rangle \\
&= d\sigma^2 \| C_1 \|_{1\to 2}^2 \sup_{X \in \mathcal{X}} \operatorname{tr}((A_2^T A_2 \circ Q) X X^T),
\end{aligned}
\tag{53}
$$

where $Q = (C_1 C_1^T)^{-1}$.

**Step 2. Bound $\mathcal{S}_2$.**

$$
\begin{aligned}
\mathcal{S}_2 &= \sup_{X \in \mathcal{X}} \mathbb{E} \langle A_2(X \bullet C_1^{-1} Z_1), A_2(C_1^{-1} Z_1 \bullet X) \rangle_{\mathrm{F}} \\
&= \sup_{X \in \mathcal{X}} \mathbb{E} \sum_{k=1}^{n} \sum_{i,j}^{d} \left( \sum_{t=1}^{n} (A_2)_{k,t} X_{t,i} (C_1^{-1} Z_1)_{t,j} \right) \left( \sum_{t=1}^{n} (A_2)_{k,t} X_{t,j} (C_1^{-1} Z_1)_{t,i} \right) \\
&= \sup_{X \in \mathcal{X}} \mathbb{E} \sum_{k=1}^{n} \sum_{i,j}^{d} \sum_{t_1,t_2}^{n} (A_2)_{k,t_1} X_{t_1,i} (C_1^{-1} Z_1)_{t_1,j} (A_2)_{k,t_2} X_{t_2,j} (C_1^{-1} Z_1)_{t_2,i} \\
&= \sup_{X \in \mathcal{X}} \mathbb{E} \sum_{k=1}^{n} \sum_{j=1}^{d} \sum_{t_1,t_2}^{n} (A_2)_{k,t_1} X_{t_1,j} (C_1^{-1} Z_1)_{t_1,j} (A_2)_{k,t_2} X_{t_2,j} (C_1^{-1} Z_1)_{t_2,j} \\
&= \frac{1}{d} \sup_{X \in \mathcal{X}} \mathbb{E} \| A_2(X \bullet C_1^{-1} Z_1) \|_{\mathrm{F}}^2 = \frac{\mathcal{S}_1}{d}
\end{aligned}
\tag{54}
$$

**Step 3. Bounding $\mathcal{S}_3$** We expand this term:

$$
\begin{aligned}
\mathcal{S}_3 &= \mathbb{E} \| A_2((C_1^{-1} Z_1) \bullet (C_1^{-1} Z_1)) \|_{\mathrm{F}}^2 \\
&= \mathbb{E} \sum_{k=1}^{n} \sum_{i,j}^{d} \left( \sum_{t=1}^{n} (A_2)_{k,t} (C_1^{-1} Z_1)_{t,i} (C_1^{-1} Z_1)_{t,j} \right)^2 \\
&= \mathbb{E} \sum_{k=1}^{n} \sum_{i,j}^{d} \sum_{t_1,t_2}^{n} (A_2)_{k,t_1} (A_2)_{k,t_2} (C_1^{-1} Z_1)_{t_1,i} (C_1^{-1} Z_1)_{t_1,j} (C_1^{-1} Z_1)_{t_2,i} (C_1^{-1} Z_1)_{t_2,j} \\
&= \mathbb{E} \sum_{k=1}^{n} \sum_{t_1,t_2}^{n} \sum_{i,j}^{d} (A_2)_{k,t_1} (A_2)_{k,t_2} (C_1^{-1} Z_1)_{t_1,i} (C_1^{-1} Z_1)_{t_1,j} (C_1^{-1} Z_1)_{t_2,i} (C_1^{-1} Z_1)_{t_2,j} \\
&= \mathbb{E} \sum_{k=1}^{n} \sum_{t_1,t_2}^{n} (A_2)_{k,t_1} (A_2)_{k,t_2} \sum_{i,j}^{d} (C_1^{-1} Z_1)_{t_1,i} (C_1^{-1} Z_1)_{t_1,j} (C_1^{-1} Z_1)_{t_2,i} (C_1^{-1} Z_1)_{t_2,j} \\
&= \sum_{k=1}^{n} \sum_{t_1,t_2}^{n} (A_2)_{k,t_1} (A_2)_{k,t_2} \sum_{i,j}^{d} \mathbb{E}(C_1^{-1} Z_1)_{t_1,i} (C_1^{-1} Z_1)_{t_1,j} (C_1^{-1} Z_1)_{t_2,i} (C_1^{-1} Z_1)_{t_2,j}.
\end{aligned}
\tag{55}
$$

We now deal with two cases. When $i = j$, we compute:

$$
\mathbb{E}(C_1^{-1}Z_1)_{t_1,j}^2(C_1^{-1}Z_1)_{t_2,j}^2
$$

$$
= \mathbb{E}\sum_{r=1}^n (C_1^{-1})_{t_1,r}^2(C_1^{-1})_{t_2,r}^2(Z_1)_{r,j}^4 + \mathbb{E}\sum_{r_1 \neq r_2}^n (C_1^{-1})_{t_1,r_1}^2(C_1^{-1})_{t_2,r_2}^2(Z_1)_{r_1,j}^2(Z_1)_{r_2,j}^2
$$

$$
+ 2\mathbb{E}\sum_{r_1 \neq r_2}^n (C_1^{-1})_{t_1,r_1}(C_1^{-1})_{t_1,r_2}(C_1^{-1})_{t_2,r_1}(C_1^{-1})_{t_2,r_2}(Z_1)_{r_1,j}^2(Z_1)_{r_2,j}^2
$$

$$
= 3\sigma^4\|C_1\|_{1\to 2}^4\sum_{r=1}^n (C_1^{-1})_{t_1,r}^2(C_1^{-1})_{t_2,r}^2 + \sigma^4\|C_1\|_{1\to 2}^4\sum_{r_1 \neq r_2}^n (C_1^{-1})_{t_1,r_1}^2(C_1^{-1})_{t_2,r_2}^2
$$

$$
+ 2\sigma^4\|C_1\|_{1\to 2}^4\sum_{r_1 \neq r_2}^n (C_1^{-1})_{t_1,r_1}(C_1^{-1})_{t_1,r_2}(C_1^{-1})_{t_2,r_1}(C_1^{-1})_{t_2,r_2}
$$

$$
= \sigma^4\|C_1\|_{1\to 2}^4 Q_{t_1,t_1}Q_{t_2,t_2} + 2\sigma^4\|C_1\|_{1\to 2}^4 Q_{t_1,t_2}^2.
$$

(56)

On the other hand, when $i \neq j$, then

$$
\mathbb{E}(C_1^{-1}Z_1)_{t_1,i}(C_1^{-1}Z_1)_{t_1,j}(C_1^{-1}Z_1)_{t_2,i}(C_1^{-1}Z_1)_{t_2,j} = \sigma^4\|C_1\|_{1\to 2}^4\left(\sum_{r=1}^n (C_1^{-1})_{t_1,r}(C_1^{-1})_{t_2,r}\right)^2
$$

$$
= \sigma^4\|C_1\|_{1\to 2}^4 Q_{t_1,t_2}^2.
$$

(57)

Using equation (56) and equation (57) in equation (55), we get

$$
\mathcal{S}_3 = \mathbb{E}\|A_2((C_1^{-1}Z_1)\bullet(C_1^{-1}Z_1))\|_F^2
$$

$$
= d\sigma^4\|C_1\|_{1\to 2}^4\sum_{k=1}^n\sum_{t_1,t_2}^n (A_2)_{k,t_1}(A_2)_{k,t_2}(Q_{t_1,t_1}Q_{t_2,t_2} + (d+1)Q_{t_1,t_2}^2)
$$

$$
= d\sigma^4\|C_1\|_{1\to 2}^4\sum_{k=1}^n\sum_{t_1,t_2}^n (A_2)_{k,t_1}(A_2)_{k,t_2}(Q_{t_1,t_1}Q_{t_2,t_2} + (d+1)Q_{t_1,t_2}^2)
$$

$$
= d\sigma^4\|C_1\|_{1\to 2}^4\,\mathrm{tr}(A_2^\top A_2 E_Q) + d(d+1)\sigma^4\|C_1\|_{1\to 2}^4\,\mathrm{tr}(A_2^\top A_2(Q\circ Q)),
$$

(58)

where $E_Q = \mathrm{diag}(Q)\,\mathrm{diag}^\top(Q)$.

Adding equation (53) to equation (55), we obtain:

$$
\sup_{X\in\mathcal{X}}\mathbb{E}\|S - \widehat{S}_{\mathrm{PP}}\|_F^2 = d(d+1)\sigma^4\|C_1\|_{1\to 2}^4\cdot\mathrm{tr}(A_2^\top A_2(Q\circ Q))
$$

$$
+ 2(d+1)\sigma^2\|C_1\|_{1\to 2}^2\cdot\sup_{X\in\mathcal{X}}\mathrm{tr}((A_2^T A_2\circ Q)XX^T)
$$

$$
+ d\sigma^4\|C_1\|_{1\to 2}^4\cdot\mathrm{tr}(A_2^\top A_2 E_Q)
$$

(59)

**Bias Correction.**

The expectation of $A_2((C_1^{-1}Z_1)\bullet(C_1^{-1}Z_1))]_{k,i,j}$ introduces a bias:

$$
[\mathbb{E}A_2((C_1^{-1}Z_1)\bullet(C_1^{-1}Z_1))]_{k,i,j} = \mathbb{E}\sum_{t=1}^n (A_2)_{k,t}\left(\sum_{r=1}^n (C_1^{-1})_{t,r}(Z_1)_{r,i}\right)\left(\sum_{r=1}^n (C_1^{-1})_{t,r}(Z_1)_{r,j}\right)
$$

$$
= \sigma^2\delta_{i,j}\|C_1\|_{1\to 2}^2\sum_{t=1}^n\sum_{r=1}^n (A_2)_{k,t}(C_1^{-1})_{t,r}^2
$$

$$
= \sigma^2\delta_{i,j}\|C_1\|_{1\to 2}^2\sum_{t=1}^n (A_2)_{k,t}Q_{t,t}.
$$

(60)

The Frobenius norm of this bias is:

$$\|\mathbb{E}A_2((C_1^{-1}Z_1)\bullet(C_1^{-1}Z_1))\|_F^2 = \sigma^4\|C_1\|_{1\to2}^4 \sum_{k=1}^{n}\sum_{j=1}^{d}\sum_{t_1,t_2}^{n}(A_2)_{k,t_1}(A_2)_{k,t_2}Q_{t_1,t_1}Q_{t_2,t_2} \tag{61}$$

$$= d\sigma^4\|C_1\|_{1\to2}^4 \operatorname{tr}(A_2^\top A_2 E_Q)$$

If we subtract this bias from the estimate, it will increase the error by the aforementioned quantity due to the Frobenius norm of the bias but will decrease the error by two scalar products with the $A_2((C_1^{-1}Z_1)\bullet(C_1^{-1}Z_1))$ term:

$$\mathbb{E}\langle A_2((C_1^{-1}Z_1)\bullet(C_1^{-1}Z_1)), \mathbb{E}A_2((C_1^{-1}Z_1)\bullet(C_1^{-1}Z_1))\rangle = \|\mathbb{E}A_2((C_1^{-1}Z_1)\bullet(C_1^{-1}Z_1))\|_F^2. \tag{62}$$

Thus, we can eliminate the last term $(d\sigma^4\|C_1\|_{1\to2}^4 \operatorname{tr}(A_2^\top A_2 E_Q))$ in the error sum via bias correction. Substituting back $\sigma = 2\zeta\sigma_{\epsilon,\delta}$, we obtain the proposed equality. $\qquad\square$

We collect some useful proposition that would be useful for our analysis.

**Proposition D.3.** *Let $V$ and $X$ be a given fixed matrix and $Z$ be a Gaussian matrix of appropriate dimension. Then*

$$\mathbb{E}_Z\|VZ\bullet VZ\|_F^2 = d(d+2)\sigma^4 \sum_{k=1}^{n}\left(\sum_{t=1}^{n}V_{k,t}^2\right)^2. \tag{63}$$

*Proof.* Recalling the Face-Splitting product, we have

$$\mathbb{E}_Z\|VZ\bullet VZ\|_F^2 = \mathbb{E}_Z\sum_{k=1}^{n}\sum_{i,j}^{d}(VZ)_{k,i}^2(VZ)_{k,j}^2 = \mathbb{E}_Z\sum_{k=1}^{n}\sum_{i,j}^{d}\left(\sum_{t=1}^{n}V_{k,t}Z_{t,i}\right)^2\left(\sum_{t=1}^{n}V_{k,t}Z_{t,j}\right)^2$$

$$= 3\sigma^4\sum_{k=1}^{n}\sum_{i=1}^{d}\sum_{t=1}^{n}V_{k,t}^4 + \sigma^4\sum_{k=1}^{n}\sum_{i\neq j}^{d}\left(\sum_{t=1}^{n}V_{k,t}^2\right)^2 + 3\sigma^4\sum_{k=1}^{n}\sum_{i=1}^{d}\sum_{j\neq\ell}^{n}V_{k,j}^2V_{k,\ell}^2$$

$$= d(d+2)\sigma^4\sum_{k=1}^{n}\left(\sum_{t=1}^{n}V_{k,t}^2\right)^2,$$

$$\tag{64}$$

This completes the proof of the proposition. $\qquad\square$

**Proposition D.4.** *Let $V$ and $X$ be a given fixed matrix and $Z$ be a Gaussian matrix of appropriate dimension. Then*

$$\mathbb{E}_Z\langle(VZ)\bullet(VX),(VX)\bullet(VZ)\rangle = \frac{1}{d}\mathbb{E}_Z\|VZ_1\bullet VX\|_F^2. \tag{65}$$

*Proof.* The result follows using the following calculation:

$$\mathbb{E}_Z\langle(VZ)\bullet(VX),(VX)\bullet(VZ)\rangle = \mathbb{E}_Z\sum_{k=1}^{n}\sum_{i,j}^{d}(VZ_1)_{k,i}(VX)_{k,j}(VZ_1)_{k,j}(VX)_{k,i}$$

$$= \mathbb{E}_Z\sum_{k=1}^{n}\left(\sum_{i=1}^{d}(VZ_1)_{k,i}(VX)_{k,i}\right)^2 \tag{66}$$

$$= \sigma^2\sum_{k=1}^{n}\sum_{t=1}^{n}V_{k,t}^2\sum_{j=1}^{d}(VX)_{k,j}^2 = \frac{1}{d}\mathbb{E}_Z\|VZ_1\bullet VX\|_F^2$$

completing the proof. $\qquad\square$

**Theorem 4.1** (Private covariance matrix estimation with JME). *Assume that all input vectors have norm at most* 1. *Let* $\widehat{\Sigma}$ *be the results of the above construction, where privacy is obtained by running JME with noise strength* $\sigma$ *and debiasing. Then it holds:*

$$\sup_{X \in \mathcal{X}} \mathbb{E}\|\Sigma - \widehat{\Sigma}\|_F^2 = (c_d d^2 + 2d + 2)\sigma^2 H_{n,1} + d(d+1)\sigma^4 H_{n,2}, \tag{11}$$

*with* $c_d$ *as in Theorem 3.3, and* $H_{n,m} := \sum_{k=1}^{n} \frac{1}{k^m}$.

*Proof.* We first recall that, if $Z \sim \mathcal{N}(\mu, \Sigma)$, then for any matrix $A$, we have $AZ \sim \mathcal{N}(A\mu, A\Sigma A^\top)$. This implies that, when $\Sigma = \mathbb{I}$ and $\mu = 0$, we have

$$\mathbb{E}_Z\|AZ\|_F^2 = \|A\|_F^2. \tag{67}$$

Recall that $\widehat{\mu} = V(X + Z_1)$ is a running mean and $\widehat{\Sigma} = V(X \bullet X + \lambda^{-1/2}Z_2) - (V(X + Z_1) \bullet V(X + Z_1))$ is a running covariance matrix, with independent noise $Z_1, Z_2 \in \mathcal{N}(0, \sigma^2)^{n \times d^2}$, the clipping norm $\zeta = 1$, and $\lambda = \lambda^* = c_d^{-1}$ as defined in equation (23). Using the associativity of Face-splitting product and the Pythogorean theorem, we have

$$\mathbb{E}_{Z_1,Z_2}\|V(X \bullet X + \lambda^{-1/2}Z_2) - (V(X + Z_1) \bullet V(X + Z_1)) - V(X \bullet X) + (VX) \bullet (VX)\|_F^2$$
$$= \mathbb{E}_{Z_1,Z_2}\|\lambda^{-1/2}VZ_2 - (VZ_1) \bullet (VX) - (VX) \bullet (VZ_1) - (VZ_1) \bullet (VZ_1))\|_F^2$$
$$= \mathbb{E}_{Z_2}\|\lambda^{-1/2}VZ_2\|_F^2 + 2\mathbb{E}_{Z_1}\|(VZ_1) \bullet (VX)\|_F^2 + \mathbb{E}_{Z_1}\|(VZ_1) \bullet (VZ_1)\|_F^2$$
$$\quad + 2\mathbb{E}_{Z_1}\langle(VZ_1) \bullet (VX), (VX) \bullet (VZ_1)\rangle$$
$$= c_d\sigma^2 d^2\|V\|_F^2 + 2\mathbb{E}_{Z_1}\|(VZ_1) \bullet (VX)\|_F^2 + \mathbb{E}_{Z_1}\|(VZ_1) \bullet (VZ_1)\|_F^2$$
$$\quad + 2\mathbb{E}_{Z_1}\langle(VZ_1) \bullet (VX), (VX) \bullet (VZ_1)\rangle \tag{68}$$

Using Proposition D.3 and Proposition D.4, we therefore have

$$\mathbb{E}_{Z_1,Z_2}\|V(X \bullet X + \lambda^{-1/2}Z_2) - (V(X + Z_1) \bullet V(X + Z_1)) - V(X \bullet X) + (VX) \bullet (VX)\|_F^2$$
$$= c_d\sigma^2 d^2\|V\|_F^2 + 2\left(1 + \frac{1}{d}\right)\mathbb{E}_{Z_1}\|(VZ_1) \bullet (VX)\|_F^2 + d(d+2)\sigma^4 \sum_{k=1}^{n}\left(\sum_{t=1}^{n} V_{k,t}^2\right)^2. \tag{69}$$

Now using the properties of $V$, we have

$$\|V\|_F^2 = \sum_{k=1}^{n}\sum_{t=1}^{n} V_{k,t}^2 = \sum_{k=1}^{n}\frac{1}{k} = H_{n,1} \quad \text{and} \quad \sum_{k=1}^{n}\left(\sum_{t=1}^{n} V_{k,t}^2\right)^2 = \sum_{k=1}^{n}\frac{1}{k^2} = H_{n,2}. \tag{70}$$

Using equation (70) in equation (69), we get

$$\mathbb{E}_{Z_1,Z_2}\|V(X \bullet X + \lambda^{-1/2}Z_2) - (V(X + Z_1) \bullet V(X + Z_1)) - V(X \bullet X) + (VX) \bullet (VX)\|_F^2$$
$$= 2\left(1 + \frac{1}{d}\right)\mathbb{E}_{Z_1}\|(VZ_1) \bullet (VX)\|_F^2 + d(d+2)\sigma^4 \sum_{k=1}^{n}\frac{1}{k^2} + 2\sigma^2 d^2 \sum_{k=1}^{n}\frac{1}{k}$$
$$= 2\left(1 + \frac{1}{d}\right)\mathbb{E}_{Z_1}\|(VZ_1) \bullet (VX)\|_F^2 + d(d+2)\sigma^4 H_{n,2} + 2\sigma^2 d^2 H_{n,1}. \tag{71}$$

Let $H_{n,c}$ denote the generalized Harmonic sum, i.e., $H_{n,c} = \sum_{i=1}^{n} i^{-c}$. Therefore, we have

$$\mathbb{E}_{Z_1,Z_2}\|V(X \bullet X + \lambda^{-1/2}Z_2) - (V(X + Z_1) \bullet V(X + Z_1)) - V(X \bullet X) + (VX) \bullet (VX)\|_F^2$$
$$= 2\left(1 + \frac{1}{d}\right)\underbrace{\mathbb{E}_{Z_1}\|(VZ_1) \bullet (VX)\|_F^2}_{S(X)} + \sigma^4 d(d+2)H_{n,2} + c_d\sigma^2 d^2 H_{n,1}. \tag{72}$$

Therefore to estimate $\sup_{X \in \mathcal{X}} \mathbb{E}\|\Sigma - \widehat{\Sigma}\|_F^2$, it suffices to estimate $\sup_{X \in \mathcal{X}} S(X)$. We do it as follows:

$$\sup_{X \in \mathcal{X}} S(X) = \sup_{X \in \mathcal{X}} \mathbb{E}_{Z_1} \|V Z_1 \bullet V X\|_F^2 = d\sigma^2 \sup_{X \in \mathcal{X}} \sum_{k=1}^{n} \frac{1}{k^3} \sum_{j=1}^{d} \left( \sum_{t=1}^{k} X_{t,j} \right)^2$$

$$= d\sigma^2 \sup_{X \in \mathcal{X}} \sum_{k=1}^{n} \frac{1}{k^3} \sum_{t_1, t_2}^{k} \langle X_{t_1,:}, X_{t_2,:} \rangle = d\sigma^2 \sum_{k=1}^{n} \frac{1}{k} = d\sigma^2 H_{n,1}, \tag{73}$$

Plugging equation (73) in equation (72), we get the bound for the biased estimate:

$$\sigma^2 (c_d d^2 + 2d + 2) H_{n,1} + \sigma^4 d(d+2) H_{n,2}. \tag{74}$$

Then we need to determine the bias term, all the first-order terms will result in $0$ as the expectation is over a zero mean distribution, so the only term that introduces the bias is

$$(\mathbb{E}_{Z_1}(V Z_1) \bullet (V Z_1))_{k,i,j} = \mathbb{E}_{Z_1}(V Z_1)_{k,i}(V Z_1)_{k,j} = \sigma^2 \delta_{i=j} \sum_{t=1}^{n} V_{k,t}^2, \tag{75}$$

where $\delta_{i=j} = \begin{cases} 1 & i = j \\ 0 & \text{otherwise} \end{cases}$ is the Dirac-delta function.

Then the unbiased approximation error is

$$\mathbb{E}_{Z_1, Z_2} \| V(X \bullet X + Z_2) - (V(X + Z_1) \bullet V(X + Z_1)) - V(X \bullet X) + (VX) \bullet (VX)$$
$$+ \mathbb{E}_{Z_1}(V Z_1) \bullet (V Z_1) \|_F^2 \tag{76}$$

We have already computed the first three terms inside the expectation. We now compute the bias term error:

$$\|\mathbb{E}_{Z_1}(V Z_1) \bullet (V Z_1)\|_F^2 = \sigma^4 \sum_{k=1}^{n} \sum_{j=1}^{d} \left( \sum_{t=1}^{n} V_{k,t}^2 \right)^2 = d\sigma^4 \sum_{k=1}^{n} \frac{1}{k^2} = d\sigma^4 H_{n,2}. \tag{77}$$

Bias reduction procedure decreases the expected error of (74) by

$$-2\mathbb{E}_{Z_1} \langle (V Z_1) \bullet (V Z_1), \mathbb{E}_{Z_1}(V Z_1) \bullet (V Z_1) \rangle + \|\mathbb{E}_{Z_1}(V Z_1) \bullet (V Z_1)\|_F^2 = d\sigma^4 \sum_{k=1}^{n} \left( \sum_{t=1}^{n} V_{k,t}^2 \right)^2$$

$$= d\sigma^4 \sum_{k=1}^{n} \frac{1}{k^2} = d\sigma^4 H_{n,2}. \tag{78}$$

Resulting in the final error of:

$$\sup_{X \in \mathcal{X}} \mathbb{E}\|\Sigma - \widehat{\Sigma}\|_F^2 = \sigma^2 (c_d d^2 + 2d + 2) \sum_{k=1}^{n} \frac{1}{k} + d(d+1)\sigma^4 \sum_{k=1}^{n} \frac{1}{k^2}$$

$$= \sigma^2 (c_d d^2 + 2d + 2) H_{n,1} + d(d+1)\sigma^4 H_{n,2}. \tag{79}$$

This completes the proof of the theorem. $\qquad\square$

**Theorem 4.2** (Private covariance matrix estimation with PP). *Assume the same setting as for Theorem 4.1. Let $\widehat{\Sigma}_{PP}$ be the result of the above construction, where privacy is obtained by running PP with noise strength $\sigma$ and debiasing. Let $S(n,d,\sigma) := d(d+1)\sigma^4 H_{n,1} - d(d+1)\sigma^4 H_{n,2} + 2(d+1)\sigma^2 H_{n,1}$. Then, for the expected error of the covariance matrix estimate it holds:*

$$S(n, d, \sigma) - 2(d+1)\sigma^2 H_{n,3} \le \sup_{X \in \mathcal{X}} \mathbb{E}\|\Sigma - \widehat{\Sigma}_{PP}\|_F^2 \le S(n, d, \sigma). \tag{12}$$

*Proof.* Let us denote the covariance matrix estimated via Post-Processing (PP) **without** bias correction as $\widehat{\Sigma}_{\text{PP}}^b$. Then, the approximation error has the following form:

$$
\begin{aligned}
\mathbb{E}\|\Sigma - \widehat{\Sigma}_{\text{PP}}^b\|_{\text{F}}^2 &= \mathbb{E}_{Z_1}\|V((X+Z_1)\bullet(X+Z_1)) - (V(X+Z_1)\bullet V(X+Z_1)) - V(X\bullet X) + (VX)\bullet(VX)\|_{\text{F}}^2 \\
&= \underbrace{\mathbb{E}_{Z_1}\|V(Z_1\bullet Z_1)\|_{\text{F}}^2}_{A_1} - 2\underbrace{\mathbb{E}_{Z_1}\langle V(Z_1\bullet Z_1),(VZ_1)\bullet(VZ_1)\rangle}_{A_2} + 2\underbrace{\mathbb{E}_{Z_1}\|V(Z_1\bullet X)\|_{\text{F}}^2}_{A_3} \\
&\quad + 2\underbrace{\mathbb{E}_{Z_1}\langle V(Z_1\bullet X),V(X\bullet Z_1)\rangle}_{A_4} - 4\underbrace{\mathbb{E}_{Z_1}\langle V(Z_1\bullet X),(VX)\bullet(VZ_1)\rangle}_{A_5} \\
&\quad - 4\underbrace{\mathbb{E}_{Z_1}\langle V(Z_1\bullet X),(VZ_1)\bullet(VX)\rangle}_{A_6} + 2\mathbb{E}_{Z_1}\|(VZ_1)\bullet(VX)\|_{\text{F}}^2 \\
&\quad + \mathbb{E}_{Z_1}\|(VZ_1)\bullet(VZ_1)\|_{\text{F}}^2 \\
&\quad + 2\mathbb{E}_{Z_1}\langle(VZ_1)\bullet(VX),(VX)\bullet(VZ_1)\rangle
\end{aligned}
\tag{80}
$$

The last three terms in the above expression evaluates to equation (72). Therefore, in what follows, we bound $A_1$ to $A_6$. **Bounding $A_1$:**

$$
\begin{aligned}
A_1 &= \mathbb{E}_{Z_1}\|V(Z_1\bullet Z_1)\|_{\text{F}}^2 = \mathbb{E}_{Z_1}\sum_{k=1}^{n}\sum_{i,j}^{d}\left(\sum_{t=1}^{n}V_{k,t}(Z_1)_{t,i}(Z_1)_{t,j}\right)^2 \\
&= 3\sigma^4\sum_{k=1}^{n}\sum_{j=1}^{d}\sum_{t=1}^{n}V_{k,t}^2 + \sigma^4\sum_{k=1}^{n}\sum_{j=1}^{d}\sum_{t_1\neq t_2}^{n}V_{k,t_1}V_{k,t_2} + \sigma^4\sum_{k=1}^{n}\sum_{i\neq j}^{d}\sum_{t=1}^{n}V_{k,t}^2 \\
&= d(d+1)\sigma^4\sum_{k=1}^{n}\sum_{t=1}^{n}V_{k,t}^2 + d\sigma^4\sum_{k=1}^{n}\left(\sum_{t=1}^{n}V_{k,t}\right)^2
\end{aligned}
\tag{81}
$$

**Bounding $A_2$:**

$$
\begin{aligned}
A_2 &= \mathbb{E}_{Z_1}\langle V(Z_1\bullet Z_1),(VZ_1)\bullet(VZ_1)\rangle \\
&= \mathbb{E}_{Z_1}\sum_{k=1}^{n}\sum_{i,j}^{d}\left(\sum_{t=1}^{n}V_{k,t}(Z_1)_{t,i}(Z_1)_{t,j}\right)\left(\sum_t V_{k,t}(Z_1)_{t,i}\right)\left(\sum_t V_{k,t}(Z_1)_{t,j}\right) \\
&= 3\sigma^4\sum_{k=1}^{n}\sum_{j=1}^{d}\sum_{t=1}^{n}V_{k,t}^3 + \sigma^4\sum_{k=1}^{n}\sum_{i\neq j}^{d}\sum_{t=1}^{n}V_{k,t}^3 + \sigma^4\sum_{k=1}^{n}\sum_{j=1}^{d}\sum_{t_1\neq t_2}^{n}V_{k,t_1}V_{k,t_2}^2 \\
&= d(d+1)\sigma^4\sum_{k=1}^{n}\sum_{t=1}^{n}V_{k,t}^3 + d\sigma^4\sum_{k=1}^{n}\left(\sum_{t=1}^{n}V_{k,t}\right)\left(\sum_{t=1}^{n}V_{k,t}^2\right)
\end{aligned}
\tag{82}
$$

**Bounding $A_3$:**

$$
A_3 = \mathbb{E}_{Z_1}\|V(Z_1\bullet X)\|_{\text{F}}^2 = \mathbb{E}_{Z_1}\sum_{k=1}^{n}\sum_{i,j}^{d}\left(\sum_{t=1}^{n}V_{k,t}(Z_1)_{t,i}X_{t,j}\right)^2 = d\sigma^2\sum_{k=1}^{n}\sum_{t=1}^{n}V_{k,t}^2\sum_{j=1}^{d}X_{t,j}^2
\tag{83}
$$

**Bounding $A_4$:**

$$
\begin{aligned}
A_4 &= \mathbb{E}_{Z_1}\langle V(Z_1\bullet X),V(X\bullet Z_1)\rangle \\
&= \mathbb{E}_{Z_1}\sum_{k=1}^{n}\sum_{i,j}^{d}\left(\sum_{t=1}^{n}V_{k,t}(Z_1)_{t,i}X_{t,j}\right)\left(\sum_{t=1}^{n}V_{k,t}(Z_1)_{t,j}X_{t,i}\right) = \sigma^2\sum_{k=1}^{n}\sum_{t=1}^{n}V_{k,t}^2\sum_{j=1}^{d}X_{t,j}^2
\end{aligned}
\tag{84}
$$

**Bounding $A_5$:**

$$
\begin{aligned}
A_5 &= \mathbb{E}_{Z_1}\langle V(Z_1 \bullet X), (VX) \bullet (VZ_1)\rangle \\
&= \mathbb{E}_{Z_1} \sum_{k=1}^{n}\sum_{i,j}^{d}\left(\sum_{t=1}^{n} V_{k,t}(Z_1)_{t,i}X_{t,j}\right)\left(\sum_{t=1}^{n} V_{k,t}X_{t,i}\right)\left(\sum_{t=1}^{n} V_{k,t}(Z_1)_{t,j}\right) \\
&= \sigma^2 \sum_{k=1}^{n}\sum_{j=1}^{d}\left(\sum_{t=1}^{n} V_{k,t}^2 X_{t,j}\right)\left(\sum_{t=1}^{n} V_{k,t}X_{t,j}\right)
\end{aligned}
\tag{85}
$$

**Bounding $A_6$:**

$$
\begin{aligned}
A_6 &= \mathbb{E}_{Z_1}\langle V(Z_1 \bullet X), (VZ_1) \bullet (VX)\rangle \\
&= \mathbb{E}_{Z_1} \sum_{k=1}^{n}\sum_{i,j}^{d}\left(\sum_{t=1}^{n} V_{k,t}(Z_1)_{t,i}X_{t,j}\right)\left(\sum_{t=1}^{n} V_{k,t}(Z_1)_{t,i}\right)\left(\sum_{t=1}^{n} V_{k,t}X_{t,j}\right) \\
&= d\sigma^2 \sum_{k=1}^{n}\sum_{j=1}^{d}\left(\sum_{t=1}^{n} V_{k,t}^2 X_{t,j}\right)\left(\sum_{t=1}^{n} V_{k,t}X_{t,j}\right)
\end{aligned}
\tag{86}
$$

Plugging equation (81) to equation (86) in equation (80), we get

$$
\begin{aligned}
\mathbb{E}\|\Sigma - \widehat{\Sigma}_{\mathrm{PP}}^{b}\|_{\mathrm{F}}^2 &= d(d+1)\sigma^4 \sum_{k=1}^{n}\sum_{t=1}^{n} V_{k,t}^2 + d\sigma^4 \sum_{k=1}^{n}\left(\sum_{t=1}^{n} V_{k,t}\right)^2 - 2d(d+1)\sigma^4 \sum_{k=1}^{n}\sum_{t=1}^{n} V_{k,t}^3 \\
&\quad - 2d\sigma^4 \sum_{k=1}^{n}\left(\sum_{t=1}^{n} V_{k,t}\right)\left(\sum_{t=1}^{n} V_{k,t}^2\right) \\
&\quad + 2(d+1)\sigma^2 \sum_{k=1}^{n}\sum_{t=1}^{n} V_{k,t}^2 \sum_{j=1}^{d} X_{t,j}^2 \\
&\quad - 4(d+1)\sigma^2 \sum_{k=1}^{n}\sum_{j=1}^{d}\left(\sum_{t=1}^{n} V_{k,t}^2 X_{t,j}\right)\left(\sum_{t=1}^{n} V_{k,t}X_{t,j}\right) \\
&\quad + 2(d+1)\sigma^2 \sum_{k=1}^{n}\sum_{t=1}^{n} V_{k,t}^2 \sum_{j=1}^{d}(VX)_{k,j}^2 + d(d+2)\sigma^4 \sum_{k=1}^{n}\left(\sum_{t=1}^{n} V_{k,t}^2\right)^2
\end{aligned}
\tag{87}
$$

Recalling that the matrix $V$ is the *averaging* workload matrix $V = (a_i^t)$ with $a_i^t = \frac{1}{t}$ for $1 \leq i \leq t$ and $a_i^t = 0$ otherwise, we get

$$
\begin{aligned}
\mathbb{E}\|\Sigma - \widehat{\Sigma}_{\mathrm{PP}}^{b}\|_{\mathrm{F}}^2 &= d(d+1)\sigma^4 \sum_{k=1}^{n}\frac{1}{k} + dn\sigma^4 - 2d(d+1)\sigma^4 \sum_{k=1}^{n}\frac{1}{k^2} - 2d\sigma^4 \sum_{k=1}^{n}\frac{1}{k} \\
&\quad + 2(d+1)\sigma^2 \sum_{k=1}^{n}\sum_{t=1}^{k}\frac{1}{k^2}\sum_{j=1}^{d} X_{t,j}^2 - 4(d+1)\sigma^2 \sum_{k=1}^{n}\sum_{j=1}^{d}\left(\sum_{t=1}^{k}\frac{1}{k^2}X_{t,j}\right)\left(\sum_{t=1}^{k}\frac{1}{k}X_{t,j}\right) \\
&\quad + 2(d+1)\sigma^2 \sum_{k=1}^{n}\frac{1}{k}\sum_{j=1}^{d}\left(\sum_{t=1}^{k}\frac{1}{k}X_{t,j}\right)^2 + d(d+2)\sigma^4 \sum_{k=1}^{n}\frac{1}{k^2} \\
&= d(d-1)\sigma^4 \sum_{k=1}^{n}\frac{1}{k} + dn\sigma^4 - d^2\sigma^4 \sum_{k=1}^{n}\frac{1}{k^2} \\
&\quad + 2(d+1)\sigma^2 \underbrace{\sum_{k=1}^{n}\frac{1}{k^2}\left[\sum_{t=1}^{k}\langle X_{t,:}, X_{t,:}\rangle - \frac{1}{k}\sum_{t_1,t_2}^{k}\langle X_{t_1,:}, X_{t_1,:}\rangle\right]}_{T_n(X)}
\end{aligned}
\tag{88}
$$

Since every term except $T_n(X)$ is independent of $X$, to compute both upper and lower bounds on $\sup_{X \in \mathcal{X}} \mathbb{E}\|\Sigma - \widehat{\Sigma}^b_{\text{PP}}\|^2_{\text{F}}$, it suffices to bound the supremum of the inner difference over $X$:

$$T_n := \sup_{X \in \mathcal{X}} T_n(X) \sup_{X \in \mathcal{X}} \sum_{k=1}^{n} \frac{1}{k^2} \left[ \sum_{t=1}^{k} \langle X_{t,:}, X_{t,:} \rangle - \frac{1}{k} \underbrace{\sum_{t_1,t_2}^{k} \langle X_{t_1,:}, X_{t_2,:} \rangle}_{=\|\sum_t X_t\|^2_2 \geq 0} \right]. \tag{89}$$

Since the double sum is non-negative, this leads to a trivial upper bound $T_n \leq \sum_{k=1}^{n} \frac{1}{k} = H_{n,1}$. For the lower bound, consider $X_i = (-1)^i e_1$. In this case, $\|X_i\|_2 = 1$, but $\|\sum_t X_t\|^2_2 \leq 1$, so $\sum_{k=1}^{n} \left( \frac{1}{k} - \frac{1}{k^3} \right) \leq T_n$. Therefore,

$$H_{n,1} - H_{n,3} \leq T_n \leq H_{n,1}. \tag{90}$$

Therefore, we have the following bounds for the approximation error **without** a bias correction:

$$\sup_{X \in \mathcal{X}} \mathbb{E}\|\Sigma - \widehat{\Sigma}^b_{\text{PP}}\|^2_{\text{F}} \leq d(d-1)\sigma^4 \sum_{k=1}^{n} \frac{1}{k} + dn\sigma^4 - d^2\sigma^4 \sum_{k=1}^{n} \frac{1}{k^2} + 2(d+1)\sigma^2 \sum_{k=1}^{n} \frac{1}{k} \tag{91}$$

$$\sup_{X \in \mathcal{X}} \mathbb{E}\|\Sigma - \widehat{\Sigma}^b_{\text{PP}}\|^2_{\text{F}} \geq d(d-1)\sigma^4 \sum_{k=1}^{n} \frac{1}{k} + dn\sigma^4 - d^2\sigma^4 \sum_{k=1}^{n} \frac{1}{k^2} + 2(d+1)\sigma^2 \sum_{k=1}^{n} \frac{1}{k} - 2(d+1)\sigma^2 \sum_{k=1}^{n} \frac{1}{k^3}, \tag{92}$$

Now we will determine the bias term; let $\delta_{i=j}$ denote the Dirac-delta function, then the bias of $\widehat{\Sigma}_{\text{PP}}$ is

$$(\mathbb{E}_{Z_1}(V(Z_1 \bullet Z_1) - \mathbb{E}_{Z_1}(VZ_1) \bullet (VZ_1))_{k,i,j} = \sigma^2 \delta_{i=j} \sum_{t=1}^{n} V_{k,t} - V^2_{k,t} \tag{93}$$

We also have the following set of equalities when $Z_1$ is a Gaussian matrix.

$$\|\mathbb{E}_{Z_1}(VZ_1) \bullet (VZ_1)\|^2_{\text{F}} = \sigma^4 \sum_{k=1}^{n} \sum_{j=1}^{d} \left( \sum_{t=1}^{n} V^2_{k,t} \right)^2 = d\sigma^4 \sum_{k=1}^{n} \frac{1}{k^2} \tag{94}$$

$$\|\mathbb{E}_{Z_1} V(Z_1 \bullet Z_1)\|^2_{\text{F}} = \sigma^4 \sum_{k=1}^{n} \sum_{j=1}^{d} \left( \sum_{t=1}^{n} V_{k,t} \right)^2 = dn\sigma^4 \tag{95}$$

$$\langle \mathbb{E}_{Z_1}(VZ_1) \bullet (VZ_1), \mathbb{E}_{Z_1} V(Z_1 \bullet Z_1) \rangle_{\text{F}} = d\sigma^4 \sum_{k=1}^{n} \left( \sum_{t=1}^{n} V^2_{k,t} \right) \left( \sum_{t=1}^{n} V_{k,t} \right) = d\sigma^4 \sum_{k=1}^{n} \frac{1}{k} \tag{96}$$

To remove the bias from the error we would need to add the following terms:

$$\|\mathbb{E}_{Z_1} V(Z_1 \bullet Z_1)\|^2_{\text{F}} + \|\mathbb{E}_{Z_1}(VZ_1) \bullet (VZ_1)\|^2_{\text{F}} - 2\langle \mathbb{E}_{Z_1}(VZ_1) \bullet (VZ_1), \mathbb{E}_{Z_1} V(Z_1 \bullet Z_1) \rangle$$
$$- 2\mathbb{E}_{Z_1} \langle (VZ_1) \bullet (VZ_1), \mathbb{E}_{Z_1}(VZ_1) \bullet (VZ_1) \rangle_{\text{F}} + 4\mathbb{E}_{Z_1} \langle (VZ_1) \bullet (VZ_1), \mathbb{E}_{Z_1} V(Z_1 \bullet Z_1) \rangle$$
$$- 2\mathbb{E}_{Z_1} \langle V(Z_1 \bullet Z_1), \mathbb{E}_{Z_1} V(Z_1 \bullet Z_1) \rangle_{\text{F}}$$
$$= -\|\mathbb{E}_{Z_1} V(Z_1 \bullet Z_1)\|^2_{\text{F}} - \|\mathbb{E}_{Z_1}(VZ_1) \bullet (VZ_1)\|^2_{\text{F}} + 2\langle \mathbb{E}_{Z_1}(VZ_1) \bullet (VZ_1), \mathbb{E}_{Z_1} V(Z_1 \bullet Z_1) \rangle$$
$$= -d\sigma^4 \sum_{k=1}^{n} \left( \sum_{t=1}^{n} V_{k,t} \right)^2 - d\sigma^4 \sum_{k=1}^{n} \left( \sum_{t=1}^{n} V^2_{k,t} \right)^2 + 2d\sigma^4 \sum_{k=1}^{n} \left( \sum_{t=1}^{n} V^2_{k,t} \right) \left( \sum_{t=1}^{n} V_{k,t} \right)$$
$$= -dn\sigma^4 - d\sigma^4 \sum_{k=1}^{n} \frac{1}{k^2} + 2d\sigma^4 \sum_{k=1}^{n} \frac{1}{k} = -dn\sigma^4 - d\sigma^4 H_{n,2} + 2d\sigma^4 H_{n,1}. \tag{97}$$

Combining everything together, we obtain:

$$\mathbb{E}\|\Sigma - \widehat{\Sigma}_{\mathrm{PP}}\|_{\mathrm{F}}^2 = d(d+1)\sigma^4 \sum_{k=1}^{n}\sum_{t=1}^{n} V_{k,t}^2 - 2d(d+1)\sigma^4 \sum_{k=1}^{n}\sum_{t=1}^{n} V_{k,t}^3$$

$$+ 2(d+1)\sigma^2 \sum_{k=1}^{n}\sum_{t=1}^{n} V_{k,t}^2 \sum_{j=1}^{d} X_{t,j}^2$$

$$- 4(d+1)\sigma^2 \sum_{k=1}^{n}\sum_{j=1}^{d} \left(\sum_{t=1}^{n} V_{k,t}^2 X_{t,j}\right)\left(\sum_{t=1}^{n} V_{k,t} X_{t,j}\right) \tag{98}$$

$$+ 2(d+1)\sigma^2 \sum_{k=1}^{n}\sum_{t=1}^{n} V_{k,t}^2 \sum_{j=1}^{d}(VX)_{k,j}^2 + d(d+1)\sigma^4 \sum_{k=1}^{n}\left(\sum_{t=1}^{n} V_{k,t}^2\right)^2$$

Therefore,

$$\sup_{X\in\mathcal{X}} \mathbb{E}\|\Sigma - \widehat{\Sigma}_{\mathrm{PP}}\|_{\mathrm{F}}^2 \leq d(d+1)\sigma^4 H_{n,1} - d(d+1)\sigma^4 H_{n,2} + 2(d+1)\sigma^2 H_{n,1} \quad \text{and} \tag{99}$$

$$\sup_{X\in\mathcal{X}} \mathbb{E}\|\Sigma - \widehat{\Sigma}_{\mathrm{PP}}\|_{\mathrm{F}}^2 \geq d(d+1)\sigma^4 H_{n,1} - d(d+1)\sigma^4 H_{n,2} + 2(d+1)\sigma^2 H_{n,1} - 2(d+1)\sigma^2 H_{n,3}, \tag{100}$$

which concludes the proof. $\qquad\square$

**Lemma A.5** (Dimension Reduction). *For any vectors $x, y \in \mathbb{R}^d$, where $d \geq 3$, there exist vectors $x', y' \in \mathbb{R}^{d-1}$ that for any $\lambda > 0$ satisfies the inequality:*

$$\|x - y\|_2^2 + \lambda\|x \circ x - y \circ y\|_2^2 \leq \|x' - y'\|_2^2 + \lambda\|x' \circ x' - y' \circ y'\|_2^2. \tag{29}$$

*Proof.* We begin by selecting indices $i$ and $j$ such that the corresponding components $x_i, x_j$ from $x$ and $y_i, y_j$ from $y$ satisfy $(x_i^2 - y_i^2)(x_j^2 - y_j^2) \geq 0$. We can always find such indices because, by the pigeonhole principle for $d \geq 3$, there will be pairs where either both $x_i^2 \geq y_i^2$ and $x_j^2 \geq y_j^2$, or both $x_i^2 \leq y_i^2$ and $x_j^2 \leq y_j^2$. We can compare the impact of these components on the sum with the values $\sqrt{x_i^2 + x_j^2}$ and $-\sqrt{y_i^2 + y_j^2}$, which correspond to vectors in a lower dimension. Consider the difference in the objective function and $f(x, i, y_i)$ defined below:

$$f(x_i, y_i) := \left(\sqrt{x_i^2 + x_j^2} + \sqrt{y_i^2 + y_j^2}\right)^2 + \lambda\left(x_i^2 + x_j^2 - y_i^2 - y_j^2\right)^2 \tag{101}$$

$$g(x_i, y_i) := (x_i - y_i)^2 + (x_j - y_j)^2 + \lambda(x_i^2 - y_i^2)^2 + \lambda(x_j^2 - y_j^2)^2. \tag{102}$$

Note that $f(x_i, y_i)$ is the objective function. Now, after algebraic manipulation, we get

$$f(x_i, y_i) - g(x_i, y_i) = 2\sqrt{x_i^2 + x_j^2}\sqrt{y_i^2 + y_j^2} + \lambda(x_i^2 - y_i^2)(x_j^2 - y_j^2) + 2x_i y_i + 2x_j y_j \tag{103}$$

$$\geq \lambda(x_i^2 - y_i^2)(x_j^2 - y_j^2) \geq 0, \tag{104}$$

where the first inequality follows from the Cauchy-Schwarz inequality and the second inequality is from the assumption that $(x_i^2 - y_i^2)(x_j^2 - y_j^2) \geq 0$.

For this lemma, $d = 2$ is indeed a special case since it is possible to find $x_1^2 > y_1^2$ and $x_2^2 < y_2^2$, for which the dimension reduction argument would not work. $\qquad\square$

**Lemma A.4.** *Consider $x, y \in \mathbb{R}^2$, and let $\lambda > 0$ then,*

$$r_2^{diag}(\lambda) = \sup_{\|x\|_2\leq 1, \|y\|_2\leq 1} \left[\|x - y\|_2^2 + \lambda\|x \circ x - y \circ y\|_2^2\right] = \begin{cases} 4, & \text{if } \lambda \leq \frac{1}{2}, \\ 2 + 2\lambda + \frac{1}{2\lambda}, & \text{if } \lambda > \frac{1}{2}. \end{cases} \tag{28}$$

*Proof.* First, we consider the effect of the signs of the components of $x$ and $y$. Multiplying both $x_i$ and $y_i$ by $-1$ does not change the value of the expression. If $x_i$ and $y_i$ have the same sign, then by flipping the sign of one of them, we increase the difference $\|x - y\|_2$, while $\|x \circ x - y \circ y\|_2$ remains unchanged. Therefore, without loss of generality, we can assume $x = (x_1, x_2)$ and $y = (-y_1, -y_2)$ for positive $x_i$ and $y_i$. Next, note that for a fixed difference $\|x - y\|_2$, the functional increases as the sum $x_1 + y_1$ or $x_2 + y_2$ increases. Thus, we can assume $\|x\|_2 = 1$, so $x_2 = \sqrt{1 - x_1^2}$. The norm of $y$, however, can be different. Therefore, we look for a solution of the form $x = (x_1, \sqrt{1 - x_1^2})$ and $y = (-y_1, -y_2)$.

We now consider the functional in the statement of the lemma in the following form:

$$\mathcal{L}_\lambda(x_1, y_1, y_2) = (x_1 + y_1)^2 + \left(\sqrt{1 - x_1^2} + y_2\right)^2 + \lambda \left(x_1^2 - y_1^2\right)^2 + \lambda \left(1 - x_1^2 - y_2^2\right)^2. \quad (105)$$

We now aim to prove that $\sup \mathcal{L}_\lambda(x_1, y_1, y_2) = r_2^{\text{diag}}(\lambda)$ in the constrained domain $y_1 \geq 0$, $y_2 \geq 0$, $y_1^2 + y_2^2 \leq 1$, $0 \leq x_1 \leq 1$. This analysis involves considering up to twenty-four different scenarios for optimization with boundaries, which, due to symmetry, can be reduced to five distinct cases.

**Case I:** $y_1^2 + y_2^2 = 1$, $y_1 > 0$, $y_2 > 0$.

We can compute $y_2 = \sqrt{1 - y_1^2}$, then the optimization functional is

$$\mathcal{L}_\lambda\left(x_1, y_1, \sqrt{1 - y_1^2}\right) = (x_1 + y_1)^2 + \left(\sqrt{1 - x_1^2} + \sqrt{1 - y_1^2}\right)^2 + \lambda \left(x_1^2 - y_1^2\right)^2 + \lambda \left(y_1^2 - x_1^2\right)^2$$
$$\quad (106)$$

$$= 2 + 2x_1 y_1 + 2\sqrt{1 - x_1^2}\sqrt{1 - y_1^2} + 2\lambda \left(x_1^2 - y_1^2\right)^2. \quad (107)$$

The cases where $x_1 = 0$ or $x_1 = 1$ with $\|y\| = 1$ are equivalent to the scenario where $\|x\| = 1$ and $y_1 = 0$, or $y_1 = 1$, which we will consider later. For now, we proceed by computing the derivatives with respect to $y_1$ and $x_1$:

$$\frac{\partial \mathcal{L}_\lambda\left(x_1, y_1, \sqrt{1 - y_1^2}\right)}{\partial y_1} = 2x_1 - \frac{2y_1 \sqrt{1 - x_1^2}}{\sqrt{1 - y_1^2}} + 4y_1 \lambda \left(y_1^2 - x_1^2\right) = 0,$$
$$\quad (108)$$
$$\frac{\partial \mathcal{L}_\lambda\left(x_1, y_1, \sqrt{1 - y_1^2}\right)}{\partial x_1} = 2y_1 - \frac{2x_1 \sqrt{1 - y_1^2}}{\sqrt{1 - x_1^2}} + 4x_1 \lambda \left(x_1^2 - y_1^2\right) = 0.$$

We transform this system by summing and subtracting the equalities:

$$(x_1 + y_1) - \frac{y_1 - y_1 x_1^2 + x_1 - x_1 y_1^2}{\sqrt{1 - y_1^2}\sqrt{1 - x_1^2}} + 4\lambda \left(x_1^3 + y_1^3\right) - 4\lambda y_1 x_1 \left(y_1 + x_1\right) = 0,$$
$$\quad (109)$$
$$(x_1 - y_1) - \frac{y_1 - y_1 x_1^2 - x_1 + x_1 y_1^2}{\sqrt{1 - y_1^2}\sqrt{1 - x_1^2}} + 4\lambda \left(y_1^3 - x_1^3\right) - 4\lambda y_1 x_1 \left(x_1 - y_1\right) = 0.$$

$x_1 + y_1 = 0$ is not a solution under the constraints we are solving. However, $x_1 = y_1$ is a solution that gives $\mathcal{L}_\lambda(x_1, x_1, \sqrt{1 - x_1^2}) = 4$ for any $\lambda$. From now on, consider $y_1 \neq x_1$; then we can divide by the difference, leading to the system:

$$1 - \frac{1 - x_1 y_1}{\sqrt{1 - y_1^2}\sqrt{1 - x_1^2}} + 4\lambda \left(x_1 - y_1\right)^2 = 0,$$
$$\quad (110)$$
$$1 + \frac{1 + x_1 y_1}{\sqrt{1 - y_1^2}\sqrt{1 - x_1^2}} - 4\lambda \left(x_1 + y_1\right)^2 = 0.$$

We consider again the sum and difference to get:

$$2 + \frac{2x_1 y_1}{\sqrt{1 - y_1^2}\sqrt{1 - x_1^2}} - 16\lambda x_1 y_1 = 0,$$
$$\quad (111)$$
$$-\frac{2}{\sqrt{1 - y_1^2}\sqrt{1 - x_1^2}} + 8\lambda \left(x_1^2 + y_1^2\right) = 0.$$

We solve it for $\lambda$ to get the following equation:

$$\frac{1}{4\left(x_1^2 + y_1^2\right)\sqrt{1 - y_1^2}\sqrt{1 - x_1^2}} = \frac{1}{8x_1 y_1} + \frac{1}{8\sqrt{1 - y_1^2}\sqrt{1 - x_1^2}}. \tag{112}$$

We transform it into the form

$$\frac{2x_1 y_1}{x_1^2 + y_1^2} - x_1 y_1 = \sqrt{1 - y_1^2 - x_1^2 + x_1^2 y_1^2}. \tag{113}$$

By squaring both sides and subtracting $x_1^2 y_1^2$, we obtain

$$\frac{4x_1^2 y_1^2}{(x_1^2 + y_1^2)^2}(1 - x_1^2 - y_1^2) = 1 - x_1^2 - y_1^2. \tag{114}$$

So, either $x_1^2 + y_1^2 = 1$, which implies $y_1 = \sqrt{1 - x_1^2}$, or $x_1^2 + y_1^2 = 2x_1 y_1$, which implies $x_1 = y_1$, which we have already discussed. We have found another potentially optimal point $y_1 = \sqrt{1 - x_1^2}$, which we will further investigate. We now consider it as a function of one variable $x_1$:

$$\mathcal{L}_\lambda\left(x_1, \sqrt{1 - x_1^2}, x_1\right) = 2\left(x_1 + \sqrt{1 - x_1^2}\right)^2 + 2\lambda\left(1 - 2x_1^2\right)^2. \tag{115}$$

Then the optimum is either $x_1 = 0$, $x_1 = 1$ with the value 4, or when the derivative is 0:

$$\frac{\partial \mathcal{L}_\lambda\left(x_1, \sqrt{1 - x_1^2}, x_1\right)}{\partial x_1} = 4\sqrt{1 - x_1^2} - \frac{4x_1^2}{\sqrt{1 - x_1^2}} - 16\lambda x_1(1 - 2x_1^2) \tag{116}$$

$$= \frac{4(1 - 2x_1^2)\left(1 - 4\lambda x_1\sqrt{1 - x_1^2}\right)}{\sqrt{1 - x_1^2}} = 0. \tag{117}$$

The first term gives $x_1 = \frac{1}{\sqrt{2}}$, which results in $\mathcal{L}_\lambda = 4$; otherwise, $4\lambda x_1\sqrt{1 - x_1^2} = 1$, which we can solve by first denoting $x_1^2$ as a new variable in the equation $-x_1^4 + x_1^2 - \frac{1}{16\lambda^2} = 0$. Using the quadratic formula, we find the optimal value $x_1^*$, which is given by:

$$x_1^* = \sqrt{\frac{1}{2} + \frac{1}{2}\sqrt{1 - \frac{1}{4\lambda^2}}}, \quad \sqrt{1 - (x_1^*)^2} = \sqrt{\frac{1}{2} - \frac{1}{2}\sqrt{1 - \frac{1}{4\lambda^2}}}. \tag{118}$$

This solution exists only when $\lambda \geq \frac{1}{2}$. Substituting this root back into the function gives us

$$\mathcal{L}_\lambda\left(x_1^*, \sqrt{1 - (x_1^*)^2}, x_1^*\right) = 2 + \underbrace{4x_1^*\sqrt{1 - (x_1^*)^2}}_{=\frac{1}{\lambda}} + 2\lambda + \underbrace{8\lambda((x_1^*)^4 - (x_1^*)^2)}_{=-\frac{1}{2\lambda}} = 2 + 2\lambda + \frac{1}{2\lambda} \geq 4, \tag{119}$$

which constitutes the function $r_2^{\text{diag}}(\lambda)$.

**Case II:** $y_1^2 + y_2^2 < 1$, $y_1, y_2 > 0$ **(Interior).**

For the optimum, it is important that $\frac{\partial \mathcal{L}_\lambda}{\partial y_1} = 0$ and $\frac{\partial \mathcal{L}_\lambda}{\partial y_2} = 0$. But the necessary condition for the maximum would be that the Hessian is negative semi-definite: $\frac{\partial^2 \mathcal{L}_\lambda}{\partial y_1^2} \leq 0$ and $\frac{\partial^2 \mathcal{L}_\lambda}{\partial y_2^2} \leq 0$. We can compute the second derivatives explicitly:

$$\frac{\partial^2 \mathcal{L}_\lambda}{\partial y_1^2} = 2 + 12\lambda y_1^2 - 4\lambda x_1^2 \le 0 \Rightarrow y_1^2 \le \frac{x_1^2}{3}. \tag{120}$$

Analogously, we can derive that $y_2^2 \le \frac{1-x_1^2}{3}$. Then substituting this into the functional, we get:

$$\mathcal{L}_\lambda(x_1, y_1, y_2) = (x_1 + y_1)^2 + \left(\sqrt{1 - x_1^2} + y_2\right)^2 + \lambda(x_1^2 - y_1^2)^2 + \lambda\left(1 - x_1^2 - y_2^2\right)^2 \tag{121}$$

$$\le \left(1 + \frac{1}{\sqrt{3}}\right)^2 + \lambda + \frac{\lambda}{9} < r_2(\lambda). \tag{122}$$

**Case III:** $y_1 = 0, 0 < y_2 < 1.$

$$\mathcal{L}_\lambda(x_1, y_1, y_2) = x_1^2 + \left(\sqrt{1 - x_1^2} + y_2\right)^2 + \lambda x_1^4 + \lambda\left(1 - x_1^2 - y_2^2\right)^2. \tag{123}$$

As a continuous function of $y_2$ on an open domain, it can reach a maximum only when the second derivative is non-positive, which leads to the same condition as in the previous case: $y_2^2 \le \frac{1-x_1^2}{3}$. Since $y_1 = 0 \le \frac{x_1^2}{3}$, this leads to the same conclusion.

**Case IV:** $y_1 = 0, y_2 = 0.$

$$\mathcal{L}_\lambda(x_1, 0, 0) = x_1^2 + 1 + \lambda x_1^4 + \lambda(1 - x_1^2)^2 \le x_1^2 + \lambda x_1^4 + \lambda(1 - x_1^2)^2 + 1 \le 1 + \lambda < r_2^{\text{diag}}(\lambda). \tag{124}$$

**Case V:** $y_1 = 1, y_2 = 0.$

$$\mathcal{L}_\lambda(x_1, 1, 0) = (x_1 + 1)^2 + 1 - x_1^2 + \lambda(1 - x_1^2)^2 + \lambda(1 - x_1^2)^2 = 2 + 2x_1 + 2\lambda(1 - x_1^2)^2. \tag{125}$$

First, for $\lambda \le \frac{1}{2}$, we have:

$$\mathcal{L}_\lambda(x_1, 1, 0) \le 2 + 2x_1 + 1 - x_1^2 \le 4. \tag{126}$$

For $\lambda > \frac{1}{2}$, we have the following inequality:

$$\mathcal{L}_\lambda(x_1, 1, 0) \le 2 + 2x_1 + 2\lambda(1 - x_1^2) = 2 + 2\lambda + 2x_1 - 2\lambda x_1^2 \le 2 + 2\lambda + \frac{1}{2\lambda}. \tag{127}$$

This concludes the proof. $\qquad\square$

**Lemma D.5** (Expected Second Moment Error with PP). *Given the private estimation of the first moment $A_1(X + C_1^{-1}Z_1)$, where $A_1 = B_1 C_1$, and the second moment $A_2(X + C_1^{-1}Z_1) \circ (X + C_1^{-1}Z_1)$, where $A_2 = B_2 C_2$, with independent noise $Z_1 \in \mathcal{N}(0, \|C_1\|_{1\to 2}^2 \sigma^2)^{n \times d}$, the clipping norm $\zeta = 1$, $d > 1$, the expected squared Frobenius norm of the estimation error for the second moment satisfies:*

$$\begin{aligned}
\sup_{X \in \mathcal{X}} \mathbb{E}_Z \|\widehat{D}_{PP} - D\|^2 :&= \sup_{X \in \mathcal{X}} \mathbb{E}\|A_2((X + C_1^{-1}Z_1) \circ (X + C_1^{-1}Z_1)) - A_2(X \circ X)\|_F^2 \\
&= 2d\sigma^4 \|C_1\|_{1\to 2}^4 \cdot \text{tr}(A_2^\top A_2 (Q \circ Q)) \\
&\quad + 4\sigma^2 \|C_1\|_{1\to 2}^2 \cdot \sup_{X \in \mathcal{X}} \text{tr}((A_2^T A_2 \circ Q)XX^T) \\
&\quad + \underbrace{d\sigma^4 \|C_1\|_{1\to 2}^4 \cdot \text{tr}(A_2^\top A_2 E_Q)}_{bias}
\end{aligned} \tag{128}$$

*where $Q = C_1^{-1} C_1^{-\top}$ and $E_Q = \text{diag}(Q)\,\text{diag}^\top(Q)$.*

*Proof.* We aim to evaluate the expected squared Frobenius norm of the error:

$$\sup_{X \in \mathcal{X}} \mathbb{E}\|A_2((X + C_1^{-1}Z_1) \circ (X + C_1^{-1}Z_1)) - A_2(X \circ X)\|_F^2$$

$$= 4 \underbrace{\sup_{X \in \mathcal{X}} \mathbb{E}\|A_2(X \circ C_1^{-1}Z_1)\|_F^2}_{S_1} + \underbrace{\mathbb{E}\|A_2((C_1^{-1}Z_1) \circ (C_1^{-1}Z_1))\|_F^2}_{S_2} \quad (129)$$

We compute those terms separately.

$$S_1 = \sup_{X \in \mathcal{X}} \mathbb{E}\|A_2(X \circ C_1^{-1}Z_1)\|_F^2 = \sup_{X \in \mathcal{X}} \mathbb{E} \sum_{k=1}^{n} \sum_{j=1}^{d} \left( \sum_{t=1}^{n} (A_2)_{k,t} X_{t,j} \sum_{r=1}^{n} (C_1^{-1})_{t,r} (Z_1)_{r,j} \right)^2$$

$$= \sup_{X \in \mathcal{X}} \sum_{k=1}^{n} \sum_{j=1}^{d} \sum_{t_1,t_2} (A_2)_{k,t_1} (A_2)_{k,t_1} X_{t_1,j} X_{t_2,j} \sum_{r=1}^{n} (C_1^{-1})_{t_1,r} (C_1^{-1})_{t_2,r} \mathbb{E}(Z_1)_{r,j}^2$$

$$= \sigma^2 \|C_1\|_{1 \to 2}^2 \sup_{X \in \mathcal{X}} \sum_{k=1}^{n} \sum_{j=1}^{d} \sum_{t_1,t_2} (A_2)_{k,t_1} (A_2)_{k,t_2} X_{t_1,j} X_{t_2,j} \sum_{r=1}^{n} (C_1^{-1})_{t_1,r} (C_1^{-1})_{t_2,r}$$

$$= \sigma^2 \|C_1\|_{1 \to 2}^2 \sup_{X \in \mathcal{X}} \sum_{t_1,t_2}^{n} \langle (A_2^\top)_{t_1}, (A_2^\top)_{t_2} \rangle \langle X_{t_1}, X_{t_2} \rangle \langle (C_1^{-1})_{t_1}, (C_1^{-1})_{t_2} \rangle$$

$$= \sigma^2 \|C_1\|_{1 \to 2}^2 \sup_{X \in \mathcal{X}} \text{tr}((A_2^T A_2 \circ Q) X X^T), \quad (130)$$

where $Q = C_1^{-1} C_1^{-T}$.

$$S_2 = \mathbb{E}\|A_2((C_1^{-1}Z_1) \circ (C_1^{-1}Z_1))\|_F^2 = \mathbb{E} \sum_{k=1}^{n} \sum_{j=1}^{d} \left( \sum_{t=1}^{n} (A_2)_{k,t} (C_1^{-1}Z_1)_{t,j}^2 \right)^2$$

$$= \mathbb{E} \sum_{k=1}^{n} \sum_{j=1}^{d} \sum_{t_1,t_2} (A_2)_{k,t_1} (A_2)_{k,t_2} (C_1^{-1}Z_1)_{t_1,j}^2 (C_1^{-1}Z_1)_{t_2,j}^2. \quad (131)$$

First, we compute:

$$\mathbb{E}(C_1^{-1}Z_1)_{t_1,j}^2 (C_1^{-1}Z_1)_{t_2,j}^2 = \mathbb{E} \sum_{r=1}^{n} (C_1^{-1})_{t_1,r}^2 (C_1^{-1})_{t_2,r}^2 (Z_1)_{r,j}^4$$

$$+ \mathbb{E} \sum_{r_1 \neq r_2}^{n} (C_1^{-1})_{t_1,r_1}^2 (C_1^{-1})_{t_2,r_2}^2 (Z_1)_{r_1,j}^2 (Z_1)_{r_2,j}^2$$

$$+ 2\mathbb{E} \sum_{r_1 \neq r_2}^{n} (C_1^{-1})_{t_1,r_1} (C_1^{-1})_{t_1,r_2} (C_1^{-1})_{t_2,r_1} (C_1^{-1})_{t_2,r_2} (Z_1)_{r_1,j}^2 (Z_1)_{r_2,j}^2$$

$$= 3\sigma^4 \|C_1\|_{1 \to 2}^4 \sum_{r=1}^{n} (C_1^{-1})_{t_1,r}^2 (C_1^{-1})_{t_2,r}^2$$

$$+ \sigma^4 \|C_1\|_{1 \to 2}^4 \sum_{r_1 \neq r_2}^{n} (C_1^{-1})_{t_1,r_1}^2 (C_1^{-1})_{t_2,r_2}^2$$

$$+ 2\sigma^4 \|C_1\|_{1 \to 2}^4 \sum_{r_1 \neq r_2}^{n} (C_1^{-1})_{t_1,r_1} (C_1^{-1})_{t_1,r_2} (C_1^{-1})_{t_2,r_1} (C_1^{-1})_{t_2,r_2}$$

$$= \sigma^4 \|C_1\|_{1 \to 2}^4 Q_{t_1,t_1} Q_{t_2,t_2} + 2\sigma^4 \|C_1\|_{1 \to 2}^4 Q_{t_1,t_2}^2. \quad (132)$$

Plugging it back, we get

$$\mathbb{E}\|A_2((C_1^{-1}Z_1) \circ (C_1^{-1}Z_1))\|_{\mathrm{F}}^2 = d\sigma^4\|C_1\|_{1\to 2}^4 \sum_{k=1}^{n}\sum_{t_1,t_2}^{n} (A_2)_{k,t_1}(A_2)_{k,t_2}(Q_{t_1,t_1}Q_{t_2,t_2} + 2Q_{t_1,t_2}^2)$$

$$= d\sigma^4\|C_1\|_{1\to 2}^4 \sum_{k=1}^{n}\sum_{t_1,t_2}^{n} (A_2)_{k,t_1}(A_2)_{k,t_2}(Q_{t_1,t_1}Q_{t_2,t_2} + 2Q_{t_1,t_2}^2)$$

$$= d\sigma^4\|C_1\|_{1\to 2}^4 \operatorname{tr}(A_2^\top A_2 E_Q) + 2d\sigma^4\|C_1\|_{1\to 2}^4 \operatorname{tr}(A_2^\top A_2(Q \circ Q)),$$
(133)

where $E_Q = \operatorname{diag}(Q)\operatorname{diag}^\top(Q)$.

Adding these terms together we obtain:

$$\sup_{X \in \mathcal{X}} \mathbb{E}_Z\|\widehat{D}_{\mathrm{PP}} - D\|^2 = 2d\sigma^4\|C_1\|_{1\to 2}^4 \cdot \operatorname{tr}(A_2^\top A_2(Q \circ Q))$$

$$+ 4\sigma^2\|C_1\|_{1\to 2}^2 \cdot \sup_{X \in \mathcal{X}} \operatorname{tr}((A_2^T A_2 \circ Q)XX^\top)$$
(134)

$$+ d\sigma^4\|C_1\|_{1\to 2}^4 \cdot \operatorname{tr}(A_2^\top A_2 E_Q)$$

**Bias Correction.**

The expectation of $A_2((C_1^{-1}Z_1) \circ (C_1^{-1}Z_1))]_{k,j}$ introduces a bias:

$$[\mathbb{E}A_2((C_1^{-1}Z_1) \circ (C_1^{-1}Z_1))]_{k,j} = \mathbb{E}\sum_{t=1}^{n}(A_2)_{k,t}\left(\sum_{r=1}^{n}(C_1^{-1})_{t,r}(Z_1)_{r,j}\right)^2$$

$$= \sigma^2\|C_1\|_{1\to 2}^2 \sum_{t=1}^{n}\sum_{r=1}^{n}(A_2)_{k,t}(C_1^{-1})_{t,r}^2$$
(135)

$$= \sigma^2\|C_1\|_{1\to 2}^2 \sum_{t=1}^{n}(A_2)_{k,t}Q_{t,t}.$$

The Frobenius norm of this bias is:

$$\|\mathbb{E}A_2((C_1^{-1}Z_1) \circ (C_1^{-1}Z_1))\|_{\mathrm{F}}^2 = \sigma^4\|C_1\|_{1\to 2}^4 \sum_{k=1}^{n}\sum_{j=1}^{d}\sum_{t_1,t_2}^{n}(A_2)_{k,t_1}(A_2)_{k,t_2}Q_{t_1,t_1}Q_{t_2,t_2}$$

$$= d\sigma^4\|C_1\|_{1\to 2}^4 \operatorname{tr}(A_2^\top A_2 E_Q)$$
(136)

If we subtract this bias from the estimate, it will increase the error by the aforementioned quantity due to the Frobenius norm of the bias but will decrease the error by two scalar products with the $A_2((C_1^{-1}Z_1) \circ (C_1^{-1}Z_1))$ term:

$$\mathbb{E}\langle A_2((C_1^{-1}Z_1) \circ (C_1^{-1}Z_1)), \mathbb{E}A_2((C_1^{-1}Z_1) \circ (C_1^{-1}Z_1))\rangle = \|\mathbb{E}A_2((C_1^{-1}Z_1) \circ (C_1^{-1}Z_1))\|_{\mathrm{F}}^2. \quad (137)$$

Thus, we can eliminate the last term $(d\sigma^4\|C_1\|_{1\to 2}^4 \operatorname{tr}(A_2^\top A_2 E_Q))$ in the error sum via bias correction.

$\square$

