# OpenReview forum: "Continual Release Moment Estimation with Differential Privacy"
_NeurIPS.cc/2025/Conference — NeurIPS 2025 poster_

### Official Review · Reviewer_TFH7 · 2025-06-20

**Clarity:** 3
**Significance:** 2
**Originality:** 3
**Rating:** 4
**Confidence:** 3

**Summary:**

The authors propose a way to release both the first and second moments in a differentially private manner, concurrently. This is also specifically for data that is being continually released, that is streaming data. A standard approach to do such a thing is allocation some budget for each of the moments. As opposed to a standard approach the proposed work reformulates the problem as a matrix and do a joint privatization using the matrix mechanism.

**Questions:**

In section 5 it is mentioned that the covariance matrices are symmetrized and projected onto the cone. Can the authors elaborate on this. How did they do the projection? If there was some clipping of eigen values, how was the clipping constant chosen?

I believe Figure 3 has an error as it states n=100 but the steps is 200.

Minor typo on page 5 toprivatize -> to privatize
Minor consistency issue, the appendix is referred to as both the "Appendix" and "appendix".

**Ethical Concerns:**

["NO or VERY MINOR ethics concerns only"]

**Final Justification:**

My original assessment of the paper was true to the paper's quality. The discussion period shed light on some questions I had but my score still seems accurate.

**Limitations:**

Yes.

**Paper Formatting Concerns:**

None.

**Quality:**

3

**Strengths And Weaknesses:**

A large endeavor in DP is being able to save budget wherever possible. Being able to release a second summary statistic at no additional cost is a great strength. As a weakness, I don't truly see the "continual release" being used thoroughly. That is, it seems to me that the theorems do not take into account what happens at time t.

---

> ### Author Rebuttal · Authors · 2025-07-30
>
> Thank you for your valuable comments. In the following, we answer your questions and will hopefully be able to address your remaining concerns.
>
>
> **Weaknesses:**
>
>
> **> “ I don't truly see the "continual release" being used thoroughly.”**
>
>
> We will clarify this in the manuscript: we follow the standard protocol for continual release DP, where the algorithm by design processes the input data in a stream, such that any moment estimate is available immediately after the corresponding data arrives. However, privacy analysis has to be done for the entire process, because privacy after step $T$, with all intermediate results available, is strictly more restrictive than privacy after any earlier step $t<T$.
> What additionally would be possible is to perform the utility analysis separately for each step $t$, not just for the final step. Our analysis can naturally be extended to intermediate steps:  we use the same noise multipliers as for the whole process, but compute the utility based on submatrices of $C_1$ and $C_2$ consisting of the first $t$ rows and columns. We will be happy to discuss this in the manuscript and restate our results for the $t$-th step.
>
>
> **Questions:**
>
>
> **> “How did they do the projection?”**
>
> We see the clipping constant as a tunable (postprocessing) parameter. As a dissimilarity measure, the KL divergence is very sensitive to small eigenvalues, which should be clipped to ~1. We observed that both Post-processing and JME need similar clipping coefficients, thus we believe this problem is intrinsic when estimating the second moment, especially from when the number of steps is small, rather than a specific artifact of JME.
>
>
> **> “Minor typo”**
>
> Thank you for spotting the typos, it should be $n=200$. We will fix this in the revised version!

---

> > ### Comment · Reviewer_TFH7 · 2025-08-02
> >
> > Thank you for your replies.
> >
> > I agree with the statement of the standard protocol for continual release. It seems to me though that with streaming data one would want to access the summaries at any time t and hence the utility at the time t seems quite relevant. I do think that, as the authors stated, this can be a direct extension of the current work.
> >
> > About the projection, I think the answer suffices but I also think this discussion is necessary. This is especially considering I'd expect the clipping to be at 0 not -1.

---

> > > ### Author Response · Authors · 2025-08-02
> > > **minor correction**
> > >
> > > Thank you for your reply. Note that in our reply we wrote "clipped to ~1" (tilde 1), meaning approximately 1, not "-1" (minus 1), which indeed would not make sense.

---

> > > > ### Comment · Reviewer_TFH7 · 2025-08-02
> > > >
> > > > Thank you for this clarification. It did seem quite obscure!

---

### Official Review · Reviewer_pJ83 · 2025-07-03

**Clarity:** 3
**Significance:** 2
**Originality:** 3
**Rating:** 4
**Confidence:** 2

**Summary:**

The paper proposes Joint Moment Estimation, a method for continual and differentially private estimation of both the first and second moments of streaming data. Unlike naive approaches that split the privacy budget between estimates, JME uses a joint sensitivity analysis to estimate the second moment without extra privacy cost—achieving what the authors call “privacy for free.” It is built upon the matrix mechanism and is adaptable to various settings (e.g., exponential weights or sliding windows). The authors prove JME's privacy and utility guarantees and show its practical benefits through applications to private Gaussian density estimation and differentially private Adam optimization, demonstrating better performance under high-privacy constraints compared to existing methods.

**Questions:**

- Figure 1: Pareto frontier behavior of \$\lambda\$-JME. Do different values of \$\lambda\$ in \$\lambda\$-JME result in different Pareto frontiers? (Do all \$\lambda\$-JME variants share the same frontier, or does the error trade-off change with \$\lambda\$?) If so, how should we interpret the dominance order of \$\lambda\_1\$-JME versus \$\lambda\_2\$-JME when \$\lambda\_1 < \lambda\_2\$?

- Could the authors discuss the computational efficiency of the proposed method, and compare it with the baselines?

- Interpretation of “privacy for free”: In line 90, the paper states that “the total sensitivity of estimating both moments...” Can this be more precisely interpreted as “the sensitivity of a linear combination of the first and second moment estimates (parameterized by \$\lambda\$)”? If so, clarifying this relationship could help make the core intuition of "privacy for free" more transparent.

- The notation \$\Vert \cdot \Vert\_{1\to 2}\$ appears before it is explained (e.g., in Algorithm 1), but its definition is given later in line 122. This makes the algorithm hard to parse on first read. It would improve readability to define or briefly explain this operator norm earlier.

**Ethical Concerns:**

["NO or VERY MINOR ethics concerns only"]

**Final Justification:**

The authors have addressed my concerns.

**Limitations:**

yes

**Quality:**

3

**Strengths And Weaknesses:**

- The paper is well-structured, with clear algorithmic descriptions.

- The paper compares JME against multiple relevant baselines (IME, CS, PP).

- For the weakness, please refer to the questions.

---

> ### Author Rebuttal · Authors · 2025-07-30
>
> We thank the reviewer for the valuable questions!
>
>
> **Questions:**
>
>
> **>  “Do different values of $\lambda$ in $\lambda$-JME result in different Pareto frontiers?”**
>
> No, this appears to be a misunderstanding. The Pareto frontier is computed by varying the trade-off parameters between first- and second-moment errors. For $\lambda$-JME, this means varying $\lambda$. For α-IME one varies the privacy split $\alpha$, and for $\tau$-CS the rescaling factor $\tau$. The trade-off preferred in practice will depend on the final application, but our analysis shows that $\lambda$-JME always is preferable, because it Pareto-dominates the other methods.
>
>
> **>  “Could the authors discuss the computational efficiency of the proposed method”?**
>
> All methods that we discuss have the same computational complexity, which is linear in the number of steps and in the output dimension (i.e quadratic in the input dimension); the difference between the methods lies in the analysis of the added noise and the resulting error. This suggests that we can achieve the same privacy level with less noise, while the practical computations remain nearly identical.
>
>
> **>  Interpretation of “privacy for free”**
>
> We are not sure what you mean by linear combinations, please correct us if we misinterpreted your comment. Note that the dimensionalities of the first and second moments differ: d and d², respectively, so it is not possible to combine them linearly. A more intuitive way to see the joint sensitivity is that we compute all intermediate first moments and all intermediate second moments, then concatenate them into a single large vector, with a scale factor multiplied to the second moment. We calculate the sensitivity of this vector (the change in its $\ell_2$-norm when changing one input point) and apply the Gaussian Mechanism to the vector. Our main insight is that for a specific choice of the scaling parameter, the sensitivity of the concatenated vector equals that of the first moment alone. We refer to this phenomenon as **“privacy for free.”**
>
>
> **> Notation**
>
> Thank you for noticing this! We will fix it in the revised version.
>
>
> We hope that our clarifications addressed your concerns, and that you will be able to support our work in the coming steps.

---

> > ### Comment · Reviewer_pJ83 · 2025-08-05
> >
> > Thanks for the clarification. I have raised the score accordingly.

---

### Official Review · Reviewer_ksfQ · 2025-07-06

**Clarity:** 3
**Significance:** 3
**Originality:** 3
**Rating:** 4
**Confidence:** 3

**Summary:**

This paper introduces a novel method called Joint Moment Estimation (JME), designed for continual, differentially private estimation of both first and second moments of a data stream. JME is tailored for continual release settings—where estimates are updated as new data arrive—and is based on the matrix mechanism for differential privacy.
The key innovation lies in a joint sensitivity analysis, which reveals a privacy regime where the second moment can be estimated without additional privacy cost relative to the first moment. This property is described as “privacy for free” for the second moment. The paper provides both theoretical guarantees and empirical results. JME achieves better accuracy than baseline methods, especially in high-privacy or small-batch settings.

**Questions:**

1. Could JME be extended to support higher-order moments (e.g., skewness, kurtosis) in a similarly privacy-efficient way?
2. How sensitive is JME’s performance to the choice of noise-shaping matrices , ? Are there principled ways to select them for arbitrary workloads?
3. Can JME be integrated with federated learning or decentralized streaming?

**Ethical Concerns:**

["NO or VERY MINOR ethics concerns only"]

**Limitations:**

The authors acknowledge that the method assumes bounded norm constraints on the data and relies on the matrix mechanism framework, which may not be straightforward for all users. Additionally, while the paper claims "privacy for free", this is bounded to the setting of joint sensitivity, and does not generalize to all DP compositions. Also, the performance is not extensively tested in large-scale or production scenarios.

**Quality:**

3

**Strengths And Weaknesses:**

Strengths:
This paper makes strong theoretical and practical contributions by providing rigorous guarantees on privacy, utility, and unbiasedness for the proposed JME algorithm. It demonstrates that second-moment estimates can be achieved with "privacy for free" through careful joint sensitivity analysis. Moment estimation being a core component in many machine learning applications further highlights JME’s real-world relevance. Empirically and theoretically, JME is shown to Pareto-dominate existing baselines such as IME, CS, and PP across utility-privacy trade-offs. The algorithm's design—leveraging matrix mechanisms and Khatri-Rao products—is both mathematically elegant and clearly presented.

Weaknesses:
The main limitations of the paper lie in its reliance on bounded input assumptions and noise-shaping matrices, which may be difficult to select or tune in practice. Additionally, the method has only been evaluated in relatively controlled or benchmark settings, lacking demonstrations in more complex, real-world machine learning pipelines. While the paper briefly mentions debiased post-processing as a baseline, it does not deeply explore its practical performance, and the theoretical complexity of the proposed approach may pose a barrier to broader adoption.

---

> ### Author Rebuttal · Authors · 2025-07-30
>
> We thank the reviewer for the valuable comments and high score!
>
>
> **Questions:**
>
>
> **> 1. “Could JME be extended to support higher-order moments?”**
>
> Thank you for the question! We believe that it is indeed possible to compute the sensitivity, at least to get an upper bound. We chose to focus on the first two moments, because these appear most commonly in practical methods, but an extension sounds like interesting future work.
>
>
> **> 2. “Are there principled ways to select them for arbitrary workloads?”**
>
> The method is sensitive to the choice of the correlation matrix, and the identity matrix may not be the optimal choice. Matrix factorization remains an active area of research, and for certain commonly used matrices, near-optimal factorizations are already known. For instance, Henzinger et al. [A Unifying Framework for Differentially Private Sums under Continual Observation, SODA 2024] showed that the matrix square root of the workload matrix achieves near-optimal performance for prefix sum workloads, exponentially decaying weights, and several others. As a general approach, we recommend numerically optimizing over the workload matrix. This has been done efficiently, for example, in McKenna [Scaling Up the Banded Matrix Factorization Mechanism for Large Scale Differentially Private ML, ICLR 2025]. We are happy to include this discussion in the revised version.
>
>
> **>3. “Can JME be integrated with federated learning or decentralized streaming?”**
>
> Yes, JME is based on the matrix factorization mechanism, which can be implemented in a federated or decentralized manner. This has been applied, e.g. in [Bienstock et al, “DMM: Distributed Matrix Mechanism for Differentially-Private Federated Learning using Packed Secret Sharing”, arXiv 2410.16161]. However, such implementations would have been orthogonal to the contribution of our paper.

---

> > ### Comment · Reviewer_ksfQ · 2025-08-07
> >
> > Thanks for the clarification from the authors.

---

### Official Review · Reviewer_wMbD · 2025-07-08

**Clarity:** 2
**Significance:** 3
**Originality:** 3
**Rating:** 4
**Confidence:** 4

**Summary:**

This paper proposed a new method named Joint Moment Estimation (JME) for continual differentially private estimation of the first and second moments of a stream of data in arbitrary number of dimensions. The method uses Gaussian noise. First and second moments are written both as matrix multiplications. Then their joint sensitivity is found, and Gaussian noise is added to both moments scaled according to the joint sensitivity. Key to reducing the noise level is to user different linear transformations and scaling factors on first and second moments to control the noise levels, hence making the joint sensitivity small. Consequentially, the second moment does not necessarily increase the noise level for the first moment. Results show improvements compared to more traditional approaches.

**Questions:**

1. This is a minor issue, but I believe the notation $||\cdot ||_{1 \rightarrow 2}$ has already been used before it becomes formally defined.
2. In the joint sensitivity (and hence noise level computation) the matrices $C_1$ and $C_2$ only comes out in the $||\cdot||_{1 \rightarrow 2}$ norm. Is that the only thing abut them that affects the noise level? Will it ever be more beneficial to use $C_1$ and $C_2$ other than the identity matrix $I$?
3. I find it a bit counter-intuitive that $c_d$ should remain a constant for every $d \geq 2$, but I didn't have to time to go through the math calculations in the appendix. Do the authors have a more high-level explanation of that?
4. The results of first moment not needing a bigger noise scale is surprising to me. I didn't check the math step by step so my confidence in this conclusion is not as high as it should be. What's the key difference between the joint sensitivity definition and the vector sensitivity in the CS method that lead to this result? For, when you concatenate the moments into a $d(d+1)$-dimension vector you can also apply the matrix $C_1$ and $C_2$, and $\lambda$ coefficient first, then take the norm and add noise according to it. Or are they roughly equivalent? Do you have any high-level intuition for that?

**Ethical Concerns:**

["NO or VERY MINOR ethics concerns only"]

**Final Justification:**

I thank the authors for their response. It addresses some of my concerns. I remain positive about accepting the paper, but still have some confusion about intuition of the theoretical approaches, practicality of the algorithm, etc. Hence I will keep my current score.

**Limitations:**

Yes.

**Paper Formatting Concerns:**

No.

**Quality:**

3

**Strengths And Weaknesses:**

Strengths:
1.  If I understand this correctly, the results in this paper indicates that by using approximate matrix multiplications and scaling on the Gaussian noise, we can add the same noise level to the first moment. The theoretical result is interesting and promising, since all privacy-budget splitting mechanisms should fail to achieve this.
2. Even when the scaling factor $\lambda$ does not give free second moment estimates, JME still pareto-dominates the IME algorithm.
3. The paper studies the continuous computation of the moments, which adds to its importance and makes it applicable for a wider variety of practical problems.
4. The algorithm is easily implementable.

Weakness:
1. The improvement over baselines in the experiments seem to be more moderate than the theory suggests. It seems that we would need $\varepsilon$ to be really small for the new algorithm to beat the baseline in DP-Adam. I wonder why that is.
2. Getting the second moment "for free" does not necessarily mean that its noise level is small, and some equations cannot be solved in closed form so it's hard to further characterize the improvement over more traditional approaches.

---

> ### Author Rebuttal · Authors · 2025-07-30
>
> We thank the reviewer for the valuable comments and high score!
>
> **Weaknesses:**
>
>
> **> 1. “The improvement over baselines in the experiments seem to be more moderate…”**
>
> The theory suggests that JME would outperform DP-Adam when the noise is greater than $1$. However, since the noise is divided by the batch size, this requires us to focus on regimes with high privacy and small batch sizes. We, however, expect JME to perform better on problems with lower dimensionality, such as numerical optimization or low-dimensional regression tasks. We will include new experiments in the revised version.
>
>
> **> 2. “...the improvement over more traditional approaches.”**
>
> Indeed, such a comparison would be interesting. However, in the paper, we consider the setting of continual release, in which classical methods for private moment estimation are not applicable.
>
>
> **Questions:**
>
>
> **> 1. Notation. Thank you, we will fix it!**
>
>
> **> 2.“Will it ever be more beneficial to use $C_1$ and $C_2$ other than the identity matrix I?”**
>
> It is true that if our goal were solely to minimize the noise per iteration, we would use identity matrices. However, our objective is to minimize the expected error and thereby maximize utility. In the utility analysis, the Frobenius norms of  $A_1 C_1^{-1}$ and $A_2 C_2^{-1}$​ also appear, which makes the choice of correlation matrices non-trivial. Henzinger et al. [A Unifying Framework for Differentially Private Sums under Continual Observation, SODA 2024] have shown that the matrix square root achieves near-optimal rates, at least for first-moment estimation.
>
>
> **> 3. Why should $c_d$ remain a constant for $d\ge2$?**
>
> We reduced the problem to an optimization over a pair of vectors $x, y \in \mathbf{R}^d$. The objective function can then be rewritten in terms of their scalar product $\langle x, y \rangle$ and their norms. In dimensions two and higher, we have independent control over the scalar product for given norms. In dimension one, this is not the case, which mainly accounts for the separation.
>
>
> **> 4. What is the key difference between JME and CS?**
>
> Thank you for the question! In JME, we reduce sensitivity by observing that the worst cases for the first and second moments differ and cannot occur simultaneously. This means that analyzing them jointly allows us to reduce the added noise. In contrast, CS analysis basically considers the worst case for each moment separately.

---

> > ### Comment · Reviewer_wMbD · 2025-08-05
> >
> > Thank you very much for addressing my comments. I've read the other reviewers' feedback and I think I have a better understanding of the paper now. I will keep my current score.

---

### Decision · Program_Chairs · 2025-09-17

**Decision:**

Accept (poster)

**Comment:**

This paper develops differentially private algorithms to simultaneously release the first and second moments from a data stream in a continual fashion in a matrix mechanism setting. They introduce an algorithm that Pareto dominates a budget-splitting approach for the two objectives of accuracy on the first moment and accuracy on the second moment, and show that in some settings, releasing the second moment does not increase the noise level for the first moment, therefore obtaining additional information “for free”. The algorithms are demonstrated on a Gaussian density estimation problem and model training with DP-Adam.

The reviewers were unanimously positive about the paper, with everyone rating it as borderline accept. They thought the paper was well structured and clear, found the result that the second moment can be obtained “for free” appealing. They found that the algorithms were practical, applicable to real problems, and showed clear improvements relative to appropriate baselines.

There were some questions about how significant the empirical gains would be in more complex settings than the relatively controlled experiments in the paper, like real ML training pipelines.

The rebuttal adequately addressed reviewer questions and clarified some misunderstandings.